# TabPack: Efficient Hyperparameter Ensembles for Tabular Deep Learning

**Yury Gorishniy** [1]  **Akim Kotelnikov** [2 1]  **Ivan Rubachev** [1 2]  **Artem Babenko** [1 2]

## Abstract

Deep learning models for supervised learning on tabular data are rapidly improving. Notably, ensembles (mixtures of multiple models) often play an important role in achieving top performance, which motivates designing *ensemble-first* systems rather than treating ensembling as an ad hoc trick. In this work, we present TabPack — a new ensembling approach that packs many base model–optimizer pairs with different hyperparameters into a single neural network and a single optimizer. The base model and optimizer hyperparameters are sampled randomly, after which all base models are trained in parallel, and the final ensemble is built on the fly during training. As a result, TabPack produces powerful ensembles in a single run, with substantial efficiency gains over traditional approaches. With its remarkable efficiency, strong performance on medium-to-large datasets, and reduced reliance on traditional hyperparameter tuning, TabPack is an appealing solution for practitioners and researchers that makes tabular DL more accessible on consumer-grade hardware and suggests a new avenue for designing better tabular deep learning systems.

## 1. Introduction

Supervised learning on tabular data is a common machine learning (ML) task in real-world applications. For a long time, the dominant approach to such tasks was gradient-boosted decision trees (GBDTs) (Chen & Guestrin, 2016; Prokhorenkova et al., 2018; Ke et al., 2017). Due to the rapid progress over recent years, modern tabular deep learning (DL) models now also demonstrate high performance and continue to improve (Holzmüller et al., 2024; Gorishniy et al., 2025; Ye et al., 2024; Erickson et al., 2025; Qu et al., 2026; Grinsztajn et al., 2026).

[1]Yandex [2]HSE University. Correspondence to: Yury Gorishniy <yurygorishniy@gmail.com>.

*Proceedings of the 43$^{rd}$ International Conference on Machine Learning*, Seoul, South Korea. PMLR 306, 2026. Copyright 2026 by the author(s).

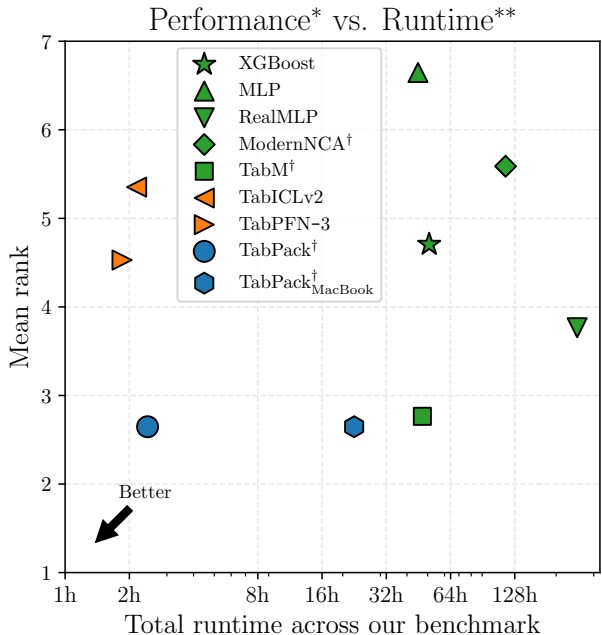

*Figure 1.* A summary of the results from Figure 6, obtained on 17 tabular datasets spanning classification and regression tasks with up to 700K+ training samples and up to 900+ features. TabPack$^{\dagger}_{\text{MacBook}}$ is evaluated on Apple M4 Pro chip with 20 GPU cores. Other methods are evaluated on NVIDIA A100 GPU.
($^{*}$) Task performance varies across benchmarks. For a more complete picture, see Figure 6 (our benchmark), subsection A.1 (large-scale industrial datasets with temporal splits) and subsection A.2 (small-to-medium datasets with IID splits).
($^{**}$) For **non-foundation** models, runtime is the hyperparameter tuning time. For **foundation** models, runtime is the inference time with default hyperparameters. For **TabPack**, runtime is the training time of a single run with default hyperparameters.

A common theme in recent work on tabular DL is the utility of ensembling, i.e., combining the predictions of multiple models. One recent example is TabArena (Erickson et al., 2025) — a benchmark where the top results are achieved by ensembles over model hyperparameters within a given model family, as well as by heterogeneous ensembles over different model families. Another example is foundation models, such as TabICL (Qu et al., 2026) and TabPFN (Grinsztajn et al., 2026), whose default prediction regime typically involves ensembling, e.g., via feature permutations. Furthermore, some *ensemble-first* tabular DL models are

beginning to emerge, such as TabM (Gorishniy et al., 2025) — a parameter-efficient ensemble of MLPs that demonstrates high performance on benchmarks.

Overall, however, the *ensemble-first* approach in tabular DL is rather in its early stages of development. For example, in the aforementioned TabArena study, the ensembling component remains largely an afterthought: the ensembles are built from base models trained independently in an ensemble-unaware manner. This post hoc approach to ensembling may not fully realize its potential in terms of task performance, and can make ensembles less efficient and less practical, particularly for production use cases. At the same time, the great success of decision-tree ensembles (Chen & Guestrin, 2016; Prokhorenkova et al., 2018; Ke et al., 2017) in classical ML remains a prominent example of how strong, efficient, and easy-to-use *ensemble-first* solutions can be, which is an important source of inspiration for this work.

We believe that the time has come to develop an *ensemble-first* tabular DL solution that incorporates the best ensemble-related practices right from the start. To this end, we introduce TabPack — a method designed from the ground up to efficiently build powerful ensembles of MLP-like models. A typical use of TabPack consists of specifying the base model and optimizer families and their hyperparameter spaces, after which a powerful ensemble is built in a single run; often, with little-to-no tuning of any hyperparameters. Thus, TabPack strikes a new appealing balance between task performance, efficiency, and sensitivity to hyperparameters, that is unavailable for traditional approaches. Overall, TabPack opens up new opportunities for building practical and powerful tabular DL systems and shows that efficiency improvements matter even in the world of small DL models.

**Main contributions.**

1. We present TabPack — a method for building ensembles of MLP-like models for tabular tasks. TabPack packs many (e.g., tens or hundreds of) base model–optimizer pairs with different hyperparameters into a single model–optimizer pair, trains all base models efficiently in a single run, and simultaneously builds an ensemble online. This workflow significantly accelerates experiment cycles for practitioners and researchers by allowing them to quickly estimate the potential of a given model–optimizer family.
2. On public benchmarks, we show that TabPack is substantially more efficient than traditional methods, while achieving competitive performance. Notably, running TabPack experiments on a modern MacBook took us less time than tuning some baselines on a discrete GPU.
3. We show that, for TabPack, the main role of hyperparameter diversity is to enable the tuning-free workflow rather than improving performance by diversifying base models.

## 2. Related Work

**Tabular machine learning.** Supervised ML on tabular data is typically addressed with classical ML models or neural networks. The strongest classical methods are gradient-boosted decision trees (GBDTs) (Chen & Guestrin, 2016; Prokhorenkova et al., 2018; Ke et al., 2017). Recent DL developments include new architectures (Gorishniy et al., 2022; 2025; Ye et al., 2024; Marton et al., 2024), training techniques (Rubachev et al., 2022; Holzmüller et al., 2024; Jeffares et al., 2023a), foundation models (Qu et al., 2026; Grinsztajn et al., 2026). Importantly, multi-layer perceptrons (MLPs) and their derivatives remain the go-to choice in modern non-foundation tabular architectures (Holzmüller et al., 2024; Gorishniy et al., 2025). Our model TabPack is an ensemble of MLP-like models with different hyperparameters. It performs on par with prior state of the art while significantly accelerating experiments through an efficient implementation and the avoidance of traditional tuning.

**Ensembles.** In machine learning, ensembling means aggregating the predictions of multiple models. In deep learning, a "deep ensemble" is a group of DL models of the same architecture trained independently (Jeffares et al., 2023b) from different initializations. Deep ensembles are known to outperform single models (Fort et al., 2020), and the diversity of ensemble members is often viewed as an important factor behind their collective performance (Allen-Zhu & Li, 2023). To reduce the costs of training multiple DL models, one can employ efficient ensembling techniques, such as packing multiple independent models into one (Laurent et al., 2023), sharing weights between ensemble members (Wen et al., 2020; Turkoglu et al., 2022), and others (Lakshminarayanan et al., 2017). Our model TabPack packs many models into one in the spirit of Packed Ensembles (Laurent et al., 2023), with the key difference that, in TabPack, the individual models differ in their hyperparameters, as do thier optimizers. Fundamentally, TabPack is also compatible with weight sharing among groups of members, although we do not employ this technique in this work for simplicity.

**Efficient hyperparameter ensembles.** To further diversify ensemble members, one can train models from the same family with different hyperparameters. As with deep ensembles in general, some studies aim to do this efficiently. One example is hyper-deep ensembles (Wenzel et al., 2020), where ensemble members are diversified by dropout rates and $L_2$ regularization coefficients. Another related example is RobOD (Ding et al., 2022), where individual models are diversified by depth and width. Our method, TabPack, further advances this idea and shows that, in principle, one can efficiently diversify any degree of freedom in models and any part of the training pipeline (e.g., optimizer hyperparameters), as long as the corresponding computations can be vectorized on modern hardware.

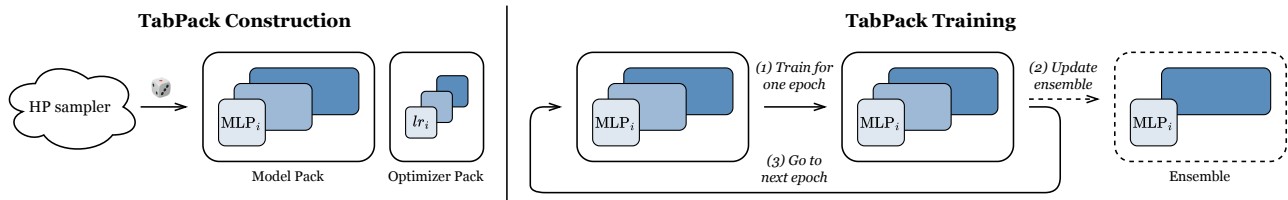

*Figure 2.* A high-level overview of TabPack. *(Left)* TabPack is constructed by packing multiple base models and optimizers with randomly sampled hyperparameters into a single model–optimizer pair. *(Right)* Training TabPack consists of training its base models independently in parallel and updating the ensemble once per epoch using intermediate base-model checkpoints. The ensemble is the main output of TabPack training.

**Ensembles in tabular ML and DL.** Common wisdom in the tabular domain is that the best performance is achieved with ensembles. For example, the recent work on the TabArena benchmark (Erickson et al., 2025) highlights hyperparameter ensembles (ensembles over models of the same family) and heterogeneous ensembles (ensembles of models from different families implemented with Auto-Gluon (Erickson et al., 2020)) as strong performers. These are examples of what we call *post hoc* or *offline* ensembles, where individual models are trained independently in an ensemble-unaware manner, and then combined by an ensemble algorithm. In contrast to post hoc ensembles, our method, TabPack represents an *ensemble-first* approach, where the goal of building an ensemble is taken into account both at the implementation level for better efficiency and at the algorithm level for better task performance, and the individual performance of base models is less of a priority.

Among tabular DL ensemble models, some methods implement differentiable tree ensembles, such as NODE (Popov et al., 2020) and GRANDE (Marton et al., 2024). However, NODE did not outperform single models in more recent studies (Gorishniy et al., 2021), and the GRANDE study is limited to classification problems, whereas our model, TabPack is not restricted to any particular task type.

Another tabular DL model closely related to this work is TabM (Gorishniy et al., 2025). It can be viewed as an ensemble-first approach: TabM is designed to represent MLP ensembles efficiently, and it performs early stopping based on the online performance of the ensemble, not of individual MLPs. Both points also apply to TabPack, however, there are several differences. *First* and most importantly, in TabM, the base MLPs share the same hyperparameters, which necessitates traditional hyperparameter tuning. By contrast, in TabPack, base-model hyperparameters are sampled randomly, allowing TabPack to build powerful ensembles with little to no tuning. *Second,* TabPack selectively includes only some base models in the ensemble, whereas TabM includes all of them. *Third,* by default, TabM uses weight sharing between base models, while TabPack does not. In principle, TabPack is compatible with weight sharing, and even without it, TabPack fits on modern GPUs.

## 3. TabPack

In this section, we present TabPack — a method for efficiently building powerful ensembles of MLP-like models for tabular data. Depending on the context, "TabPack" can denote either the model itself or the overall system including the model, optimizer, and other components.

**Overview.** Figure 2 provides a high-level illustration of TabPack. The key technical elements of TabPack include:

1. A model architecture design in the style of Packed Ensembles (Laurent et al., 2023) for *faster training* of base models in parallel (subsection 3.2, subsection 3.3, subsection 3.4).
2. Simultaneous training of all base models for *better task performance* through online ensemble construction (subsection 3.5) and early stopping based on ensemble performance as in TabM (Gorishniy et al., 2025) (subsection 3.6).
3. Randomly sampled hyperparameters for the base models and optimizers making TabPack usable with *little to no tuning* (subsection 3.7).

The rest of this section describes TabPack in detail.

### 3.1. Preliminaries

**Notation.** This work considers supervised regression and classification problems on tabular data. For a given object in a dataset, we use $x$ and $y$ to denote its features and label, respectively, and $\hat{y}$ to denote the prediction of $y$ by an ML model. We use $d$ to denote dimensionalities and $W_i$ to denote linear-layer weights in the $i$-th layer of a neural network. All notation may be used with additional subscript or superscript labels depending on the context.

**Benchmarks.** Our benchmark is derived from Gorishniy et al. (2025) and includes eight industrial datasets with temporal splits from the challenging TabReD benchmark (Rubachev et al., 2025) and nine datasets from other sources. This gives us 17 medium-to-large regression and classification datasets of diverse domains and sizes ranging from 10K to 1M+ objects and from 8 to 900+ features. See Appendix E for details.

**Experiment setup.** We follow the experiment setup of Gorishniy et al. (2025) and describe it in detail in subsection F.1. In particular, we split each dataset into training, validation, and test sets, where the validation part is used for things like hyperparameter tuning and early stopping, and the test part is used to compute the final metrics. The metric definitions, namely ranks and relative improvements over MLP, are also inherited from Gorishniy et al. (2025).

## 3.2. Packed Ensembles: a Quick Recap

Efficiently constructing ensembles from base MLP-like models is the central topic of our work. To train base models as fast as possible, we use a design in the spirit of Packed Ensembles (Laurent et al., 2023). For our purposes, it is enough to see this method as a way to pack multiple models of the same architecture into one by simply (1) stacking the model parameters and inputs across a new pack dimension, and (2) applying all models to all inputs in parallel by relying on the broadcasting mechanism available in mainstream DL frameworks. To give an idea of the speedups from this design, we compare the inference throughput of tabular MLPs in Figure 3. The figure covers three inference regimes corresponding to three potential approaches to training many individual models on one GPU: sequential, parallel and packed training. The results indicate that packing is dramatically more efficient than the alternatives.

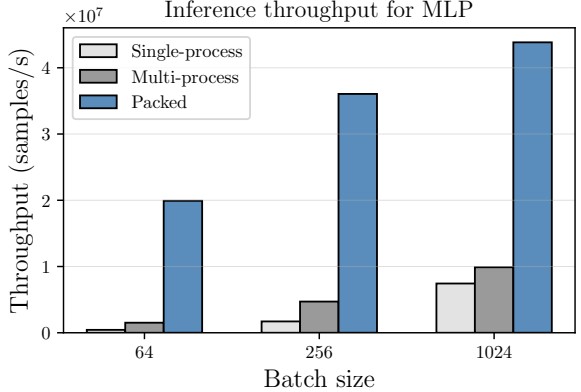

*Figure 3.* Inference throughput for an MLP of depth 3 and width 384 on one NVIDIA A100 GPU with different batch sizes in three regimes: *(1)* single-process, *(2)* multi-process, where multiple processes run in parallel using the same GPU (the process count is tuned to show the full potential of this approach), and *(3)* packed (64 MLPs are stacked into one and applied to 64 stacked batches in parallel).

## 3.3. Model Pack

As a model, TabPack represents $m$ MLP-like models, potentially with different hyperparameters, applied in parallel to $m$ inputs in a single forward pass. We refer to this pattern as a *model pack*, and to the individual models as *base*

*models*. For simplicity, we use a plain MLP (a sequence of Linear–ReLU–Dropout blocks) as the base architecture, so the varying hyperparameters include depth, width, and dropout rates. In principle, one can extend TabPack with residual connections (He et al., 2016), normalizations (Ioffe & Szegedy, 2015; Ba et al., 2016), and other custom layers, and vary base models by hyperparameters and presence or absence of these modules.

**Implementation.** To pack heterogeneous base MLPs into one, we increase the width and depth of each base MLP to the maximum values across the pack and stack their parameters across a new pack dimension, as illustrated in Figure 4. This allows applying all $m$ MLPs to $m$ input batches in parallel using batched matrix multiplications (e.g., `torch.bmm` in PyTorch (Paszke et al., 2019)). To recover the behavior of the original base MLPs, we do the following:

- For base models narrower than the maximum width, the corresponding parts of intermediate representations are zeroed out as shown on the left side of Figure 5.
- For base models shallower than the maximum depth, only the necessary number of leading linear layers are applied, and the remaining deeper layers are skipped.
- Varying dropout rates are implemented by sampling $m$ dropout masks with $m$ different rates.

The described approach can be efficiently implemented in all mainstream DL frameworks using appropriate vectorized operations. We note that to realize the ensemble potential of TabPack, one can usually use smaller maximum widths and depths than in traditional single models. This, together with vectorization, alleviates the overhead from performing all matrix multiplications in $\mathbb{R}^{d_{\max}}$ instead of $\mathbb{R}^{d_i}$ ($d_i \leq d_{\max}$) and from shallower models "waiting" for deeper models during the forward pass.

## 3.4. Optimizer Pack

To support diverse training-related hyperparameters across base models, we implement *optimizer packs*, analogously to model packs. In optimizer packs, base optimizers differ in their learning rates, weight decays, and potentially other hyperparameters, such as momentum coefficients. The right side of Figure 5 shows this mechanism for the vanilla SGD; in practice, we use more advanced optimizers, as discussed in subsection 3.8. For an optimizer pack, the state consists of individual optimizer states stacked along a new pack dimension, where tensor shapes in each individual state follow those of the deepest and widest MLP.

## 3.5. Ensemble

Including all base models of TabPack in the final ensemble is not viable, since random hyperparameter sampling for base models (discussed in subsection 3.7) inevitably makes some

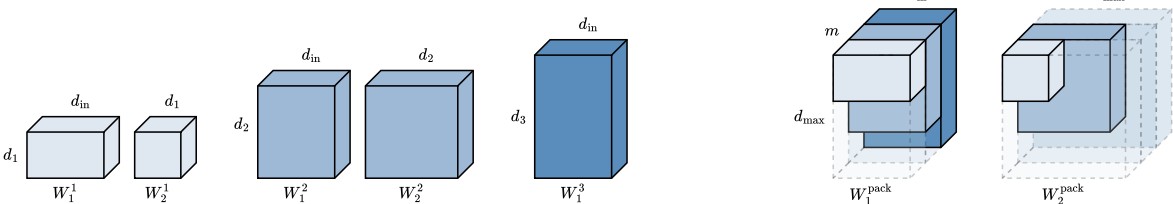

*Figure 4.* Packing multiple heterogeneous MLPs into one. *(Left)* Weights of $m = 3$ MLP backbones of different widths ($d_1 < d_2 < d_3$) and depths (2, 2, 1). Biases are omitted for simplicity. $W_j^i$ denotes the weights of the $j$-th layer in the $i$-th MLP. *(Right)* Weights of the same $m = 3$ MLPs as on the left, but now packed in one set of weights.

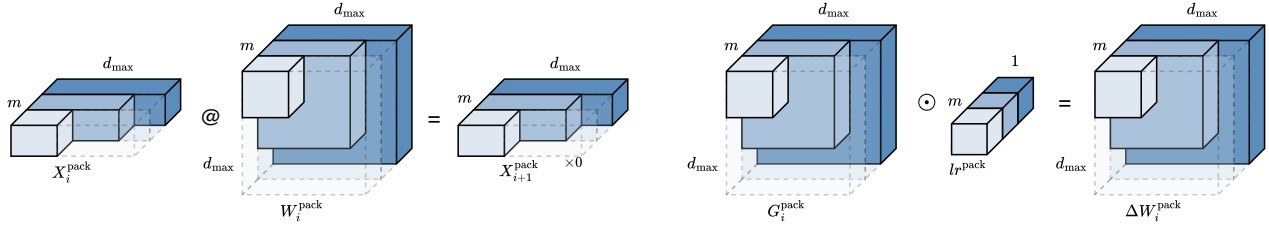

*Figure 5. (Left)* A forward pass for a pack of linear layers. The input $X$ consists of $m$ object representations padded with zeros to $d_{\max}$ and stacked along the pack dimension. The remaining notation follows Figure 4. First, a batched matrix multiplication applies, where all layers are applied in parallel to different input representations, as indicated by colors. Then, the output is masked according to the actual layer dimensions, as indicated by the "$\times 0$" label. *(Right)* The weight-update computation for a pack of SGD optimizers, where $G$ denotes gradients. In practice, TabPack uses more advanced optimizers.

base models poorly suited as ensemble members. Thus, the ensemble must be constructed from base models selectively. We consider two approaches for this: *offline* and *online*, both described later in this section.

**Greedy ensembling: a quick recap.** Before presenting the offline and online approaches to ensembling, we briefly describe the greedy ensembling algorithm used by both as the core primitive for selecting ensemble members from a given pool of base models, also called *candidates*. We define greedy ensembling as follows:

- Start by selecting the best base model according to the *ensemble score* on the validation set. Unless stated otherwise, we use the target task metric as the ensemble score. In practice, depending on the dataset, using the training loss as the ensemble score is sometimes a better choice.
- In a loop, on each iteration, add to the ensemble the base model from the pool that improves the ensemble performance the most on the validation set (the individual model performance is ignored). Base models from the pool are added to the ensemble without replacement.
- Stop when the specified ensemble size $m_{\text{ens}}$ is reached or when no further improvement is possible.

While more advanced ensembling algorithms exist (see Caruana et al. (2004) as an entry point to the topic), we found the greedy algorithm to be a decent simple baseline.

**Offline ensemble.** The simple *offline* approach is to first train all base models independently (using TabPack as "infrastructure" for faster training) and then select the ensemble members. Modulo TabPack's efficient implementation, this is the strategy used in Erickson et al. (2025).

The key strength of the offline approach is its simplicity. At the same time, it has two important limitations. First, intermediate base-model states remain unused, significantly limiting the candidate pool for ensemble construction, which can result in worse performance. Second, there is no way to stop the base-model training based on ensemble performance, which again can hurt performance and also makes the total runtime equal to that of the slowest-to-train base model.

**Online ensemble.** With the above in mind, TabPack uses the *online* approach by default. Generally, this means updating the ensemble continuously during training using intermediate states of the base models. In this work, we rebuild the ensemble from scratch once per epoch. At a given epoch, the pool of candidates consists of the latest base model states and the ensemble members from the previous epoch. Because the previous ensemble members are added to the pool, multiple instances of the same base model from different epochs can become ensemble members, since such instances are treated as independent candidates.

### 3.6. Training

**Base models.** The base models of TabPack are trained in parallel on different batch sequences. Thus, one training batch for TabPack consists of $m$ base batches of size $B$ stacked along the new pack dimension. The models are trained independently: their *losses* are aggregated, not their predictions.

**Early stopping for the ensemble.** We stop training after $p_{ens} + 1$ consecutive epochs without improvement in ensemble performance on the validation set.

**Early stopping for the base models.** To further accelerate TabPack training, we apply early stopping to base models as well, so that more compute is allocated to the still-improving base models. Specifically, if a base model does not improve its validation performance for $p_{base} + 1$ consecutive epochs, it is removed from the pack.

### 3.7. Hyperparameters

Hyperparameters for the base models of TabPack are sampled randomly from user-defined hyperparameter spaces. Thus, from the perspective of TabPack, these spaces are themselves hyperparameters. Other hyperparameters of TabPack include the number of base models $m$, early-stopping patience $p_{base}$ for base models and $p_{ens}$ for the ensemble, standard training-related options (e.g., a batch size), and ensembling-related options (e.g., the maximum ensemble size $m_{ens}$).

For simplicity, in this work we never tune TabPack's hyperparameters and always use $m = 64$, $p_{base} = 16$, $p_{ens} = 32$, $m_{ens} = 32$ (to match the TabM baseline (Gorishniy et al., 2025)), and the base hyperparameter spaces specified in subsection F.3. In particular, for simplicity, we use a fixed base-model width instead of sampling diverse values. Overall, every performance number reported for TabPack is a result of a single run (modulo multi-seed runs for evaluation purposes).

### 3.8. Practical Notes

**Base model family.** We observe that the base model family has a significant impact on the ensemble performance, and techniques aimed at improving individual model performance remain highly relevant for the ensemble performance of TabPack. In particular, we find that feature embeddings (Gorishniy et al., 2022) bring substantial benefits, and therefore recommend using them by default. In this work, we use a variant of periodic embeddings (see Appendix B for details), and diversify their hyperparameters similarly to those of the MLP backbone.

Overall, it seems that the ensembling mechanism behind TabPack efficiently realizes the potential of a given model family, but cannot fully compensate for the fundamental limitations of the base model family. Thus, developing better individual models remains a worthy research direction.

**Optimizer family.** In line with Gorishniy et al. (2026), we found Muon (Jordan et al., 2024) to consistently outperform AdamW (Loshchilov & Hutter, 2019) both for TabPack and for prior baselines (see subsection A.4 for details). Thus, we adopt Muon as the default optimizer for all methods in all experiments.

**Limitations** are discussed in Appendix C.

## 4. Experiments

In this section, we compare TabPack against tabular ML and DL baselines on public benchmarks.

### 4.1. Models

We use the following baselines: XGBoost (Chen & Guestrin, 2016) — a powerful GBDT implementation; plain MLP, as in Gorishniy et al. (2025); ModernNCA (Ye et al., 2024) — a modern retrieval-based model; RealMLP (Holzmüller et al., 2024) — an advanced MLP-like architecture combined with a specific training recipe (the recipe is unique to RealMLP and not used by TabPack by other baselines); TabM (Gorishniy et al., 2025) — a parameter-efficient ensemble of MLPs; TabICLv2 (Qu et al., 2026) and TabPFN-3 (Grinsztajn et al., 2026) — modern tabular foundation models (TFMs). See Appendix F for implementation details.

**MLP$_{Ens}$.** Additionally, we introduce MLP$_{Ens}$ — essentially a multi-process implementation of TabPack with offline ensembling, inspired by the "Tuned + Ensemble" evaluation regime in TabArena (Erickson et al., 2025). That is, all base models with randomly sampled hyperparameters are trained independently in $N_p$ concurrent processes on one GPU, and the ensemble is constructed afterwards. In the experiments, we set $N_p$ to the value that reveals the full efficiency potential of MLP$_{Ens}$. Although we expect MLP$_{Ens}$ to lag behind TabPack in both effectiveness and efficiency, this baseline has its strengths. In particular, it does not require reimplementing standard layers and optimizers, and it allows all ensemble-related logic to be concentrated in a separate base-model-agnostic pipeline. Thus, we believe it is worth quantifying the difference between TabPack and its multi-process version to better understand the trade-offs.

† **Feature embeddings.** Given the significant effect of feature embeddings (Gorishniy et al., 2022) on task performance, we explicitly mark models that use feature embeddings with †: TabPack$^{†}$, MLP$^{†}$, etc. For a fair comparison, we use the same feature embeddings for all models, namely, a variant of periodic embeddings (see Appendix B).

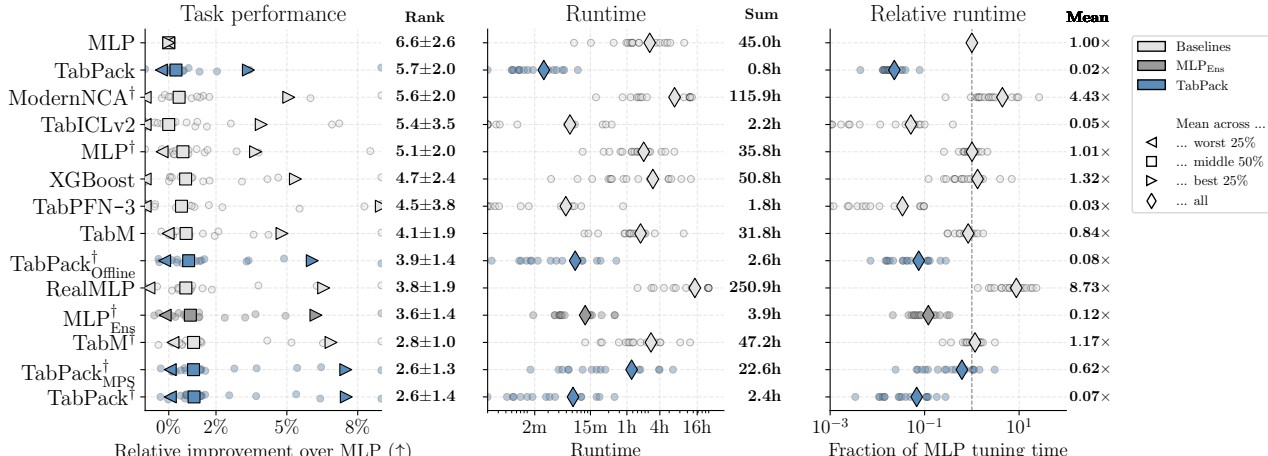

*Figure 6. (Left)* Task performance of tabular models, on the 17 datasets from Table 5. The models are sorted by mean rank. Each dot represents the result on one dataset, and the triangle–square–triangle markers denote mean values across disjoint dataset groups (the groups vary between models) as explained in the legend. *(Middle)* The absolute runtime on the same datasets as in the left plot. For non-foundation models, this is the hyperparameter tuning time. For foundation models, this is the inference time. For TabPack and MLP$_{Ens}$, this is the time of a single run, since there is no tuning. *(Right)* Same as the middle plot, but all values are computed relative to the tuning time of MLP.

**TabPack.** We evaluate TabPack both without (TabPack) and with (TabPack$^\dagger$) feature embeddings. Additional variants of TabPack include TabPack$_{Offline}$ — the version of TabPack with offline ensembling (see subsection 3.5), and TabPack$_{MPS}$— TabPack trained on a MacBook without a discrete GPU (see Appendix D).

### 4.2. Evaluation Protocols

**Evaluation protocol for baselines.** Most baselines undergo traditional hyperparameter tuning followed by multi-seed evaluation of the tuned configuration, as in Gorishniy et al. (2025). The only exceptions are foundation models, which are evaluated with their default hyperparameters and with the validation set merged into the training set.

**Evaluation protocol for TabPack.** The somewhat unconventional workflow of TabPack requires a dedicated protocol for multi-seed evaluation. There are two possible approaches:

- (Optimistic) Run TabPack $N_{seeds}$ times with different random seeds and report performance and runtime aggregated over the seeds.
- (Conservative) First, perform the "main" run of TabPack once. Suppose that $M_{ens}$ base models were selected for the final ensemble; in this work, $M_{ens} \leq m_{ens} = 32 < 64 = m$). Then, rerun TabPack $N_{seeds}$ times with different random seeds using only the $M_{ens}$ selected base models. Report the task performance aggregated over the random seeds, and the runtime of the main run.

The conservative protocol naturally results in worse

TabPack performance than the optimistic protocol, because its secondary runs use fewer base models. Nevertheless, we use the *conservative* protocol to better align with the tune-once routine of traditional models.

**Evaluation protocol for MLP$_{Ens}$.** As with TabPack, we use the *conservative* protocol for MLP$_{Ens}$, which means running MLP$_{Ens}$ once, then rerunning it $N_{seeds}$ times with the subset of base models selected during the main run, and reporting aggregated task performance of the secondary runs and the runtime of the main run.

### 4.3. Task Performance And Runtime

**Results.** We visualize the results in Figure 6 and summarize them as follows:

- TabPack shows strong performance and substantial speedups over the traditional experiment workflow. Remarkably, TabPack enables faster experiment cycles on a modern MacBook (represented by TabPack$_{MPS}$) than some prior methods on an enterprise-grade GPU, which is an NVIDIA A100 in our case.
- The online approach to ensembling is superior to the offline one (represented by TabPack$_{Offline}$ and MLP$_{Ens}$), although the latter can still be of interest due to its simplicity, decent performance, and good efficiency.
- Performance-wise, TabPack is more stable than some of the baselines, as reflected by the "Mean across worst 25%" performance markers on the left size of the figure.
- Additionally, in subsection A.3, we discuss the preliminary observation that TabPack performs especially well on large regression tasks.

**Impact.** In light of the above results, we highlight the following implications of TabPack for tabular DL:

- **TabPack offers a competitive new baseline with fast experiment cycles** for practitioners and researchers.
- **TabPack makes modern tabular DL more accessible** in terms of both hyperparameter tuning and hardware requirements.
- **TabPack offers a new evaluation regime:** a given method can be evaluated not only in isolation in the traditional manner, but also as the base method in TabPack or a similar framework.

**Main takeaway:** TabPack is an effective and highly efficient approach to solving ML problems on tabular data.

### 4.4. Comparison across Time Budgets

We now compare the task performance of tabular models under varying time budgets. Specifically, on a given dataset and for a given time budget, we measure the following:

- For TabPack, the performance achieved in a single run when training is stopped after the given time budget.
- For $MLP_{Ens}$, the performance of the ensemble greedily built from the fully trained base models available after the given time budget.
- For traditional methods, the performance of the best hyperparameter configuration found when hyperparameter tuning is stopped after the given time budget.

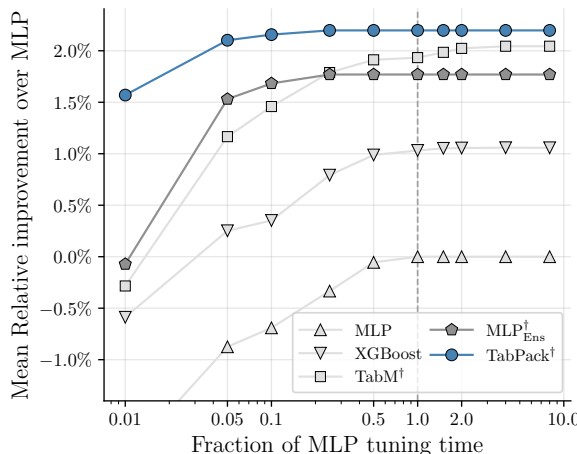

*Figure 7.* Performance of tabular models as a function of the time budget, averaged over the 17 datasets from Table 5 and over three runs with different random seeds. See subsection 4.4 for details.

The results in Figure 7 indicate that TabPack is a strong model across a wide range of time budgets, further highlighting its potential as a go-to solution for practice and fast research experiments. In particular, we note how quickly TabPack realizes most of its potential. Additionally, we again highlight $MLP_{Ens}$ as a decent baseline.

**Main takeaway:** TabPack is an effective tabular DL solution across various time budgets.

### 4.5. Task Performance on Other Benchmarks

After the joint evaluation of performance and runtime in the previous sections, we additionally evaluate task performance alone on other benchmarks, where full-fledged efficiency measurements are difficult and/or expensive to obtain. We consider two additional benchmarks:

- **Large datasets with 1M+ training objects**, where TabPack outperforms baselines on four out of five datasets. See subsection A.1.
- **Small-to-medium datasets with IID splits from TabArena** (Erickson et al., 2025), a benchmark that is more challenging for TabPack due to its smaller scale, where TabPack performs competitively overall, especially in the tuning-free regime. See subsection A.1.

## 5. Analysis

### 5.1. The Role of Hyperparameter Diversity

Random hyperparameter sampling enables the tuning-free workflow of TabPack. However, beyond its efficiency and convenience benefits, it is unclear whether diverse hyperparameters are actually important for task performance. To this end, we additionally evaluate $TabPack_{SameHP}$: TabPack with all base models having the **S**ame hyperparameters tuned in a traditional manner on each dataset separately (see subsection F.2 for implementation details). By analogy, the workflow $TabPack_{SameHP}$ is the same as that of TabM, where all base models also share the same hyperparameters, which must be tuned. The results in Table 1 show that the tuning slightly improves the performance at the cost of noticeable increase of the total runtime. Thus, random hyperparameter sampling is a reasonable default strategy; its main applied role is reducing the need for tuning rather than improving task performance by diversifying the base models.

*Table 1.* The performance and runtime on the same datasets as in Figure 6 of three TabPack variants described in subsection 5.1. For $TabPack^{\dagger}_{SameHP}$, the runtime is the hyperparameter tuning time across all datasets, while for the rest it is the time of a single run.

| Model | Rank | Runtime |
|---|---|---|
| $TabPack^{\dagger}$ | $1.6 \pm 0.6$ | 2.4h |
| $TabPack^{\dagger}_{DiverseWidths}$ | $1.6 \pm 0.5$ | 3.5h |
| $TabPack^{\dagger}_{SameHP}$ | $1.4 \pm 0.7$ | 65.5h |

A related question is how our choice to use the same width for all base models affects performance. To answer this, we test TabPack variant where the base backbone width is

sampled from $\mathrm{Uniform}[64, 512]$, and we denote this variant as $\mathrm{TabPack_{DiverseWidths}}$. As shown in Table 1, sampling varying model widths results in comparable performance with an increase in runtime caused by the higher maximum model width. However, it is possible that the constant-width approach is competitive due to other details of our setup that implicitly compensate for the lack of diversity in model widths. We also note that varying the width requires patching public Muon implementations to work correctly.

### 5.2. A Closer Look at the Final Ensembles

Now, we take a closer look at the ensembles produced by TabPack. We will analyze the runs of TabPack[†] reported in Figure 6.

**Ensemble size.** Across all datasets, the mean number of base models in the final ensemble is $16 \pm 9$ (as a reminder, the maximum allowed ensemble size is 32). For comparison, the ensemble size of TabM used in the original paper is 32. Thus, while TabPack uses more base models during its single run, the final ensemble sizes are comparable to those of prior work. This, in particular, ensures practical inference efficiency, as shown in subsection A.5.

**Hyperparameter space utilization.** Recall that the hyperparameters for base models of TabPack are sampled randomly from a predefined space. To estimate to what extent this space is utilized by the final ensemble on a given dataset, for each hyperparameter, we divide its ensemble range (the difference between the maximum and the minimum values in the ensemble) by the total allowed range, and report this metric in Table 2. The results indicate a non-trivial hyperparameter diversity between the ensemble members.

*Table 2.* Hyperparameter space utilization, as defined in subsection 5.2, aggregated over the runs of TabPack[†] from Figure 6. $d_{\mathrm{emb}}$ and $\sigma$ are feature embeddings hyperparameters (see Appendix B).

| Hyperparameter | Space utilization |
|---|---|
| Depth | $0.88 \pm 0.27$ |
| Dropout | $0.77 \pm 0.28$ |
| $d_{\mathrm{emb}}$ | $0.90 \pm 0.16$ |
| $\sigma$ | $0.81 \pm 0.19$ |
| Learning rate (Muon) | $0.72 \pm 0.24$ |
| Learning rate (AdamW) | $0.75 \pm 0.26$ |
| Weight decay | $0.68 \pm 0.16$ |

### 5.3. Hyperparameter Exploration

Due to the limited presence of TabPack-like ensemble-first methods in the tabular DL landscape, it is unclear how the key model-related hyperparameters affect the task performance of the final ensemble. To provide some intuition on this aspect, we vary the main architectural hyperparameters

of TabPack and report the results in Figure 8. From the results, we conclude that the configuration used in this work can serve as a starting point for future work. However, we note that our results reflect TabPack's behavior on specific datasets and under specific experiment protocol. For example, richer base hyperparameter spaces may require higher values of $m$ to fully realize TabPack's potential.

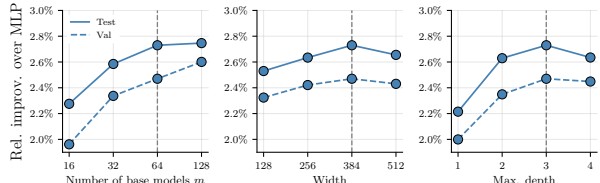

*Figure 8.* Task performance of TabPack[†] depending on different hyperparameters, averaged over the 17 from Table 5. The vertical dashed lines correspond to the values used throughout the paper.

## 6. Conclusion & Future Work

In this work, we presented TabPack— an efficient ensemble of MLP-like models for solving machine learning problems on tabular data. The key feature of TabPack is its ability to efficiently produce powerful ensembles with little to no tuning, enabling significantly faster experimental cycles. This makes TabPack an appealing baseline for practitioners and researchers, as well as an effective tabular DL framework for plugging-in more advanced base methods.

Examples of potential directions for future work include advancing the online ensemble algorithm behind TabPack, reducing the reliance of TabPack on the validation set, developing better base models for their use in TabPack. Additionally, we separately highlight the opportunity to extend the TabPack-style hyperparameter diversification to other parts of the training pipeline, such as loss functions and data preprocessing, as well as the opportunity to vary base models not only by hyperparameters but also by architectural elements.

### Impact Statement

This paper presents work whose goal is to advance the field of Machine Learning. There are many potential societal consequences of our work, none which we feel must be specifically highlighted here.

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

# A. Additional Analysis and Results

## A.1. Evaluation on Datasets with $\geq 1$M training objects

In this section, we compare tabular models on datasets with more than one million training objects. Specifically, we use five industrial datasets with temporal splits summarized in Table 6. Due to the long runtimes, for the baselines, we reuse the hyperparameters tuned in the main experiments in Figure 6 on subsampled versions of the same datasets. For TabPack, we use the single-run tuning-free workflow as in the main text with the base model patience reduced to $p_{\text{base}} = 4$ and the ensemble patience reduced to $p_{\text{ens}} = 8$. The results in Table 3 indicate that TabPack successfully scales to large datasets outperforming baselines on four out of five tasks.

*Table 3.* The task performance on five large datasets from the TabReD benchmark (Rubachev et al., 2025). The metrics are ROC-AUC for Homecredit Default and RMSE for the rest of the datasets, averaged over three random seeds.

| Dataset | Training set size | #Features | TabPack$^{\dagger}$ | TabM$^{\dagger}$ | MLP$^{\dagger}$ | XGBoost | TabPFN$-3$ |
|---|---|---|---|---|---|---|---|
| Homecredit default | 1.1M | 696 | 0.8693 | **0.8737** | 0.8689 | 0.8728 | 0.8494 |
| Delivery eta | 11.2M | 223 | **0.5404** | 0.5410 | 0.5422 | 0.5413 | OOM |
| Weather | 13.6M | 103 | **1.3899** | 1.4065 | 1.4567 | 1.4158 | OOM |
| Maps routing | 6.4M | 986 | **0.1572** | 0.1595 | 0.1586 | 0.1602 | OOM |
| Cooking time | 9.1M | 192 | **0.4758** | 0.4780 | 0.4780 | 0.4782 | OOM |

## A.2. Evaluation on TabArena

In this section, we compare tabular models on the TabArena benchmark (Erickson et al., 2025) — a benchmark consisting of of small-to-medium-sized datasets with IID random splits. This may be a more challenging setup for TabPack, since TabArena datasets have validation sets of smaller absolute and relative sizes than those in our benchmark, and, as explained in Appendix C, this may be problematic for TabPack. Nevertheless, as shown in Figure 9 and Figure 10, TabPack demonstrates competitive performance on TabArena. In particular, in the "Default" regime (the blue bars), TabPack confidently outperforms all models except foundation models, which are known to be strong on TabArena-like benchmarks.. **Implementation note:** in this section, we use the training loss as the ensemble score for the greedy ensembling algorithm powering TabPack, as we observed this variant to perform better on TabArena classification tasks.

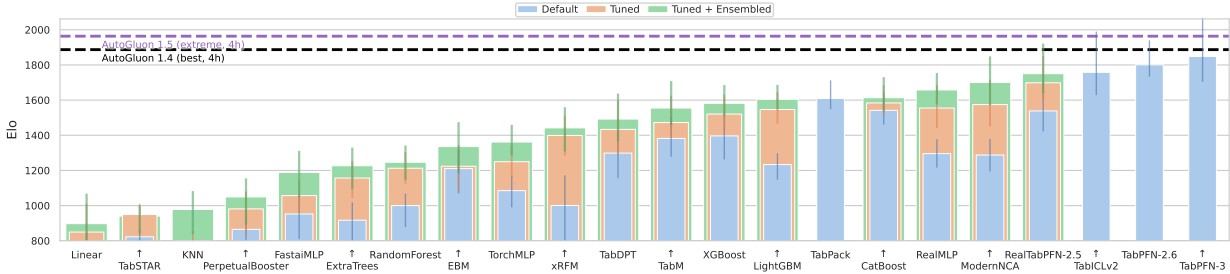

*Figure 9.* Results on *medium-sized* datasets (at least 10K objects in total) from the TabArena-Lite benchmark (Erickson et al., 2025).

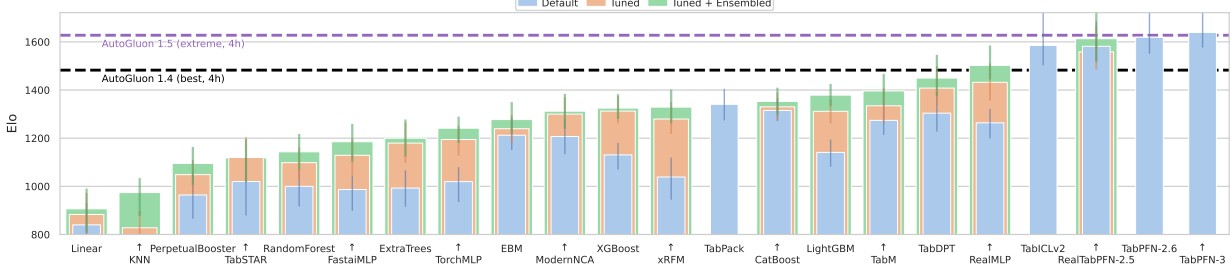

*Figure 10.* Results on *small-sized* datasets (less than 10K objects in total) from the TabArena-Lite benchmark (Erickson et al., 2025).

### A.3. A Note on (Large) Regression Tasks

A closer look at the results in subsection A.1 and Table G reveals an interesting pattern: TabPack is not just competitive, but actually noticeably superior to competitors specifically on large regression tasks. Due to the limited number of such datasets, it is hard to tell whether this is a general phenomenon or a coincidence. Plus, specifically in subsection A.1, the baselines did not undergo the full-fledged hyperparameter tuning due to the high cost of experiments.

As a thought experiment, let's assume that the strong performance of TabPack on large regression tasks is a general pattern. Then, the underlying reason for this pattern may be the combination of the following factors:

- By its nature, the greedy ensembling algorithm used by TabPack gradually refines the ensemble prediction.
- The online approach to ensembling used by TabPack greatly expands the set of ensemble member candidates considered by TabPack, which further strengthens TabPack's ability to gradually improve the ensemble during training.
- Unlike more "discrete" classification metrics, such as accuracy, typical regression metrics, such as RMSE or $R^2$, are more rewarding for gradual prediction improvements. In turn, using more "continuous" metrics, such as cross-entropy, to guide ensemble construction on classification tasks creates the discrepancy between the ensemble score and the actual task metric (although it can be beneficial sometimes; for example, see subsection A.2). Regression metrics such as RMSE or $R^2$ are free from this problem.
- On average, the larger the dataset, the better its validation set serves as a proxy for its test set. This helps TabPack to generalize well to the test set despite using the validation set to guide ensemble construction.

### A.4. Muon vs. AdamW

As mentioned in the main text, we find Muon (Jordan et al., 2024) to consistently outperform AdamW (Loshchilov & Hutter, 2019) as the optimizer for TabPack. Specifically, when compared under exactly the same experiment protocol as in subsection 4.3 across all 17 datasets from Table 5:

- For TabPack, the win/tie/loss counts of Muon against AdamW are 9/3/5.
- For TabPack[†], the win/tie/loss counts of Muon against AdamW are 13/3/1.

Where "win" and "loss" simply mean ranks 1 and 2, respectively, and "tie" means that both methods have rank 1.

### A.5. Inference Efficiency

In this section, we quickly estimate how TabPack compares to prior work in terms of inference efficiency. We start with two observations:

- The TabM paper (Gorishniy et al., 2025) contains detailed measurements of the inference throughput of tabular models on GPU and CPU, showing that TabM exhibits practical inference characteristics suitable for many real-world use cases.
- In this paper, the ensembles produced by TabPack are almost surely more inference-efficient than TabM, because: (1) as mentioned in subsection 5.2, the final ensemble size of TabPack is smaller than that of TabM; (2) the base model width used in TabPack is 384, while in TabM it is tuned in $\mathrm{Uniform}[64, 1024]$; (3) the maximum base model depth of TabPack is 3, while in TabM it is tuned in $\mathrm{Uniform}[1, 5]$.

With the above in mind, it is reasonable to expect that TabPack is at least as efficient as TabM. To validate this intuition, we perform a quick experiment and report the results in Figure 11. The results are in line with the expectations and establish TabPack as a practical solution for real-world usage.

### A.6. Memory Consumption during Training

In Table 4, we report the memory consumption of TabPack during training. The results indicate that TabPack training will often fit in a consumer-grade GPU. That said, for large enough number of base models and large enough number of features in a dataset, TabPack may require hardware with more than 12GB memory to run.

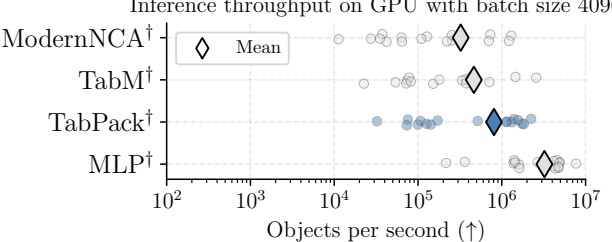

*Figure 11.* Inference throughput on NVIDIA A100 on 17 datasets from Table 5

*Table 4.* Peak memory usage in GB during training averaged over 17 datasets from Table 5 depending on the number of base models.

| Num base models: | 1 | 2 | 4 | 8 | 16 | 32 | 64 | 128 |
|---|---|---|---|---|---|---|---|---|
| TabM[†] | 0.3 | 0.4 | 0.5 | 0.7 | 1.2 | 2.1 | 4.0 | 7.7 |
| TabPack[†] | 0.3 | 0.4 | 0.6 | 0.9 | 1.6 | 3.0 | 5.8 | 11.3 |

## B. Feature Embeddings

**Background.** An embedding for continuous features (Gorishniy et al., 2022) is a mapping $f_{\text{emb}} : \mathbb{R} \to \mathbb{R}^{d_{\text{emb}}}$ applied to each continuous feature in isolation *before* mixing the features in the main backbone, where $d_{\text{emb}}$ is the embedding size. A simple example of a non-linear feature embedding is $f_{\text{emb}}(x_i) = \text{ReLU}(w_i \cdot x + b_i)$, where $i$ is the feature index, and $w_i, b_i \in \mathbb{R}^{d}_{\text{emb}}$ are trainable parameters (not shared between features). When feature embeddings are used with MLP-like models, the backbone takes the input consisting of all feature embeddings concatenated in one flat vector.

**New periodic embeddings.** In this work, we introduce a new variation of "periodic embeddings", i.e. embeddings using periodic activation functions under the hood. Prior work includes PLR embeddings from Gorishniy et al. (2022) and PBLD embeddings from Holzmüller et al. (2024). Our variant adopts certain design elements from prior work and improves efficiency noticeably by avoiding hidden layers.

Our periodic embedding $f_{\text{periodic}} : \mathbb{R} \to \mathbb{R}^{d_{\text{emb}}}$ is formally defined as follows:

$$f_{\text{periodic}}(x_i) = \alpha_i \cdot [x_i, \cos(w_i \cdot x + b_i)] + \beta_i$$

where $\cdot$ denotes element-wise multiplication, $+$ denotes element-wise addition, $[\ldots]$ denotes concatenation, $i$ is the feature index, $w_i \in \mathbb{R}^{d_{\text{emb}}-1}$ is initialized from $N(0, \sigma)$ ($\sigma \in \mathbb{R}$ is a hyperparameter), $b_i \in \mathbb{R}^{d_{\text{emb}}-1}$ is initialized with zeros, and $\alpha_i, \beta_i$ are also initialized with zeros.

**Notation.** In this work, all models marked with † use our periodic embeddings described above.

## C. Limitations

We highlight the following limitations of TabPack:

- In our current implementation, TabPack relies on the validation set for both ensemble construction and early stopping. We hypothesize that overfitting to the validation set may become an issue if the validation set is small or there is a significant distribution shift between the validation and test sets.
- TabPack naturally requires more GPU memory for training compared to traditional single models. We report memory usage in subsection A.6. In GPU-memory-constrained setups, one can reduce the maximum base model dimensions (in particular, feature embedding sizes) and/or the number of base models to alleviate the issue. Another strategy is a fallback to offline ensembles: run TabPack multiple times with the reduced number of base models, and build an ensemble using all base models afterwards.
- To extend TabPack with new architectural elements and optimizers, one has to implement their packed versions instead of simply reusing existing implementations. We hope that our code base will serve as a helpful example for implementing new packed building blocks.

## D. Hardware

All experiments, modulo the runs of TabPack$_{\text{MPS}}$, were conducted in the single-GPU regime using NVIDIA A100 80GB.

For TabPack$_{\text{MPS}}$, we used a 16-inch MacBook Pro with the Apple M4 Pro chip with 10 performance CPU cores, 4 efficiency CPU cores, 20 GPU cores and 48GB of unified RAM. The compute backend was `torch.device("mps")`, so the integrated GPU was used as the main compute device.

## E. Datasets

Table 5 summarizes properties of the datasets used in most experiments throughout the paper, while Table 6 summarizes properties of the datasets used specifically in subsection A.1. Note that some of the TabReD datasets are presented in both tables. This is because they have two official versions, full and the subsampled, both introduced in the TabReD paper. For subsection A.1, we prepare variants of these datasets with full training sets, but samsampled validation and test sets, to make the results comparable between the two versions.

*Table 5.* Extended properties of datasets used in experiments throughput the paper. Here, "# Train", "# Val", "# Test" denotes the size of the corresponding dataset split; similarly, "# Num", "# Bin", "# Cat" denotes the number of numerical, binary, and categorical features, respectively.

| Name | Source | # Train | # Val | # Test | # Num | # Bin | # Cat | Task Type | Batch Size |
|---|---|---|---|---|---|---|---|---|---|
| Churn Modelling | | 6 400 | 1 600 | 2 000 | 7 | 3 | 1 | Binclass | 128 |
| California Housing | | 13 209 | 3 303 | 4 128 | 8 | 0 | 0 | Regression | 256 |
| House 16H | | 14 581 | 3 646 | 4 557 | 16 | 0 | 0 | Regression | 256 |
| Adult | TABM | 26 048 | 6 513 | 16 281 | 6 | 1 | 8 | Binclass | 256 |
| Diamond | (Gorishniy et al., 2025) | 34 521 | 8 631 | 10 788 | 6 | 0 | 3 | Regression | 512 |
| Otto Group Products | | 39 601 | 9 901 | 12 376 | 93 | 0 | 0 | Multiclass | 512 |
| Higgs Small | | 62 751 | 15 688 | 19 610 | 28 | 0 | 0 | Binclass | 512 |
| Black Friday | | 106 764 | 26 692 | 33 365 | 4 | 1 | 4 | Regression | 512 |
| Microsoft | | 723 412 | 235 259 | 241 521 | 131 | 5 | 0 | Regression | 1024 |
| Sberbank Housing | | 18 847 | 4 827 | 4 647 | 365 | 17 | 10 | Regression | 256 |
| Ecom Offers | | 109 341 | 24 261 | 26 455 | 113 | 6 | 0 | Binclass | 1024 |
| Maps Routing | | 160 019 | 59 975 | 59 951 | 984 | 0 | 2 | Regression | 1024 |
| Homesite Insurance | TABRED | 224 320 | 20 138 | 16 295 | 253 | 23 | 23 | Binclass | 1024 |
| Cooking Time | (Rubachev et al., 2025) | 227 087 | 51 251 | 41 648 | 186 | 3 | 3 | Regression | 1024 |
| Homecredit Default | | 267 645 | 58 018 | 56 001 | 612 | 2 | 82 | Binclass | 1024 |
| Delivery ETA | | 279 415 | 34 174 | 36 927 | 221 | 1 | 1 | Regression | 1024 |
| Weather | | 340 596 | 42 359 | 40 840 | 100 | 3 | 0 | Regression | 1024 |

*Table 6.* Extended properties of the datasets used in subsection A.1. See Appendix E for details.

| Name | Source | # Train | Other properties |
|---|---|---|---|
| Maps Routing | | 6 408 198 | |
| Cooking Time | TABRED | 9 096 168 | |
| Homecredit Default | with full training sets | 1 070 762 | Same as in Table 5 |
| Delivery ETA | (Rubachev et al., 2025) | 11 151 785 | |
| Weather | | 13 625 138 | |

## F. Implementation details

### F.1. Experiment Setup

We mostly rely on the experiment setup from Gorishniy et al. (2025). As such, a significant portion of the below text is copied from that work, with some modifications related to our study.

**Data preprocessing.** For each dataset, for all DL-based solutions, the same preprocessing was used for fair comparison. For numeric features, by default, we used a modified version of the quantile normalization from the Scikit-learn package (Pedregosa et al., 2011), with rare exceptions when it turned out to be detrimental (for such datasets, we used the standard

normalization or no normalization). For categorical features, we used one-hot encoding. Binary features (i.e. the ones that take only two distinct values) are treated as categorical features. For the datasets from TabReD(Rubachev et al., 2025), we follow the data preprocessing from the original paper.

**Training baseline neural networks.** For DL-based algorithms, we minimize cross-entropy for classification problems and mean squared error for regression problems. We use the Muon optimizer Jordan et al. (2024) for the main (rectangular) weights of linear layers in the main backbones, and the AdamW optimizer (Loshchilov & Hutter, 2019) for everything else. We do not apply learning rate schedules. We do not use data augmentations. For each dataset, we used a predefined dataset-specific batch size provided in Table 5. We continue training until there are `patience` + 1 consecutive epochs without improvements on the validation set; we set `patience` = 16 for DL models.

**Hyperparameter tuning.** In most cases, hyperparameter tuning is performed with the TPE sampler (typically, 50-100 iterations) from the Optuna package (Akiba et al., 2019). Hyperparameter tuning spaces for most models are provided in individual sections below.

**Evaluation.** On a given dataset, for a given model, the tuned hyperparameters are evaluated under multiple (in most cases, 5 or 10 random seeds. The mean test metric and its standard deviation over these random seeds are then used to compare algorithms.

### F.2. Implementation Details of subsection 5.1

To obtain TabPack$^{\dagger}_{\text{SameHP}}$, we use the standard hyperparameter tuning pipeline as for DL baselines in this work, i.e. it the standard tuning process using the TPE sampler from Optuna (Akiba et al., 2019). In this tuning process, the hyperparameter tuning space is exactly the same as the one used for random hyperparameter sampling in TabPack.

### F.3. TabPack

The hyperparameter space used to sample base model hyperparameters in TabPack$^{\dagger}$ is provided in Table 7. For TabPack, it is the same, but without feature embedding hyperparameters and the maximum allowed depth is 4 instead of 3.

*Table 7.* The hyperparameter sampling space for TabPack$^{\dagger}$ base models

| Parameter | Distribution |
|---|---|
| # layers | $\text{UniformInt}[1, 3]$ |
| Width (hidden size) | $384$ |
| Dropout rate | $\{0.0, \text{Uniform}[0.0, 0.5]\}$ |
| $d_{\text{emb}}$ | $\text{UniformInt}[8, 32]$ (step 4) |
| $\sigma$ | $\text{LogUniform}[1e\text{-}2, 10]$ |
| Muon learning rate | $\text{LogUniform}[1e\text{-}3, 1e\text{-}1]$ |
| AdamW learning rate | $\text{LogUniform}[1e\text{-}4, 5e\text{-}3]$ |
| Weight decay | $\text{LogUniform}[1e\text{-}3, 1]$ |

### F.4. MLP

All MLP variants use the Muon optimizer for main network parameters and AdamW for remaining parameters (e.g., embeddings, biases). We train all MLP models in float32 precision. Table 8 and Table 9 provide the hyperparameter tuning spaces for MLP and MLP$^{\dagger}$ with the new periodic embeddings, respectively.

### F.5. TabM

All TabM variants use the Muon optimizer for main network parameters and AdamW for the remaining parameters. We use $k = 32$ for all TabM experiments. We train TabM baselines in BFloat16 precision. Table 10 and Table 11 provide the hyperparameter tuning spaces for TabM and TabM$^{\dagger}$ with the new embeddings, respectively.

*Table 8.* The hyperparameter tuning space for MLP.

| Parameter | Distribution |
|---|---|
| # layers | $\mathrm{UniformInt}[1, 6]$ |
| Width (hidden size) | $\mathrm{UniformInt}[64, 1024]$ |
| Dropout rate | $\{0.0, \mathrm{Uniform}[0.0, 0.5]\}$ |
| Muon learning rate | $\mathrm{LogUniform}[1e\text{-}4, 3e\text{-}2]$ |
| AdamW learning rate | $\mathrm{LogUniform}[3e\text{-}5, 1e\text{-}3]$ |
| Weight decay | $\mathrm{LogUniform}[1e\text{-}3, 1]$ |
| # Tuning iterations | 100 |

*Table 9.* The hyperparameter tuning space for MLP with embeddings.

| Parameter | Distribution |
|---|---|
| # layers | $\mathrm{UniformInt}[1, 5]$ |
| Width (hidden size) | $\mathrm{UniformInt}[64, 1024]$ |
| Dropout rate | $\{0.0, \mathrm{Uniform}[0.0, 0.5]\}$ |
| $d_{\mathrm{emb}}$ | $\mathrm{UniformInt}[8, 32]$ (step 4) |
| $\sigma$ | $\mathrm{LogUniform}[1e\text{-}2, 10]$ |
| Muon learning rate | $\mathrm{LogUniform}[1e\text{-}4, 3e\text{-}2]$ |
| AdamW learning rate | $\mathrm{LogUniform}[3e\text{-}5, 1e\text{-}3]$ |
| Weight decay | $\mathrm{LogUniform}[1e\text{-}3, 1]$ |
| # Tuning iterations | 100 |

*Table 10.* The hyperparameter tuning space for TabM. Here, (B) = {Black Friday, Microsoft, and all TabReD datasets}. (A) contains all other datasets.

| Parameter | Distribution or Value |
|---|---|
| $k$ | 32 |
| # layers | $\mathrm{UniformInt}[1, 5]$ |
| Width (hidden size) | $\mathrm{UniformInt}[64, 1024]$ |
| Dropout rate | $\{0.0, \mathrm{Uniform}[0.0, 0.5]\}$ |
| Muon learning rate | $\mathrm{LogUniform}[1e\text{-}3, 1e\text{-}1]$ |
| AdamW learning rate | $\mathrm{LogUniform}[1e\text{-}4, 5e\text{-}3]$ |
| Weight decay | $\mathrm{LogUniform}[1e\text{-}3, 1]$ |
| # Tuning iterations | (A) 100 (B) 50 |

### F.6. ModernNCA

We adapted the official implementation from Ye et al. (2024), with modifications according to our experiment protocol. We use the new modified periodic embeddings and train in float32 precision.

Table 12 provides the hyperparameter tuning space for ModernNCA[†]. When tuning the ModernNCA model on larger datasets (this includes all TabReD datasets without Ecom Offers and Sberbank Housing and Microsoft) we set a timeout on the tuning procedure at 12 hours.

*Table 11.* The hyperparameter tuning space for TabM with embeddings. Here, (B) = {Black Friday, Microsoft, and all the TabReD datasets.}(A) contains all other datasets.

| Parameter | Distribution or Value |
|---|---|
| $k$ | 32 |
| # layers | UniformInt$[1, 4]$ |
| Width (hidden size) | UniformInt$[64, 1024]$ |
| Dropout rate | $\{0.0, \text{Uniform}[0.0, 0.5]\}$ |
| $d_{\text{emb}}$ | UniformInt$[8, 32]$ (step 4) |
| $\sigma$ | LogUniform$[1e\text{-}2, 10]$ |
| Muon learning rate | LogUniform$[1e\text{-}3, 1e\text{-}1]$ |
| AdamW learning rate | LogUniform$[1e\text{-}4, 5e\text{-}3]$ |
| Weight decay | LogUniform$[1e\text{-}3, 1]$ |
| # Tuning iterations | (A) 100 (B) 50 |

*Table 12.* The hyperparameter tuning space for ModernNCA$^{\dagger}$. Here, (B) = {Microsoft and TabReD datasets except Ecom Offers and Sberbank Housing}. (A) contains all other datasets.

| Parameter | Distribution |
|---|---|
| # blocks | UniformInt$[0, 2]$ |
| dim | UniformInt$[64, 1024]$ |
| $d_{\text{block}}$ | UniformInt$[64, 1024]$ |
| Dropout rate | Uniform$[0.0, 0.5]$ |
| Sample rate | Uniform$[0.05, 0.6]$ |
| $d_{\text{emb}}$ | (A) UniformInt$[8, 32]$ (B) UniformInt$[4, 16]$ (step 4) |
| $\sigma$ | LogUniform$[1e\text{-}2, 10]$ |
| Muon learning rate | LogUniform$[1e\text{-}4, 3e\text{-}2]$ |
| AdamW learning rate | LogUniform$[3e\text{-}5, 1e\text{-}3]$ |
| Weight decay | LogUniform$[1e\text{-}3, 1]$ |
| # Tuning iterations | (A) 100 (B) 50 |

## F.7. RealMLP

We used the `tabarena-new` preset from pytabkit[1] and TPE sampler instead of random one in the original implementation. We also adapted Muon optimizer for RealMLP and tuned Muon learning rate with LogUniform$[0.02, 0.3]$ space.

## F.8. TabICL-V2

We used the official `tabicl` Python package[2]. We merged the validation set to the training set. We used ordinal encoding for categorical features. All other hyperparameters were kept at their default values.

## F.9. TabPFN-3

We used the official `tabpfn` Python package[3]. We merged the validation set to the training set. We used ordinal encoding for categorical features and pass them using `categorical_feature_indices`. All other hyperparameters were kept

---

[1] https://github.com/dholzmueller/pytabkit/tree/c126ea5

[2] https://github.com/soda-inria/tabicl

[3] https://github.com/PriorLabs/TabPFN

at their default values.

### F.10. XGBoost

Table 13 provides the hyperparameter tuning space for XGBoost.

*Table 13.* The hyperparameter tuning space for XGBoost.

| Parameter | Distribution or Value |
|---|---|
| n_estimators | 4000 |
| max_depth | UniformInt$[3, 14]$ |
| learning_rate | LogUniform$[1e\text{-}3, 1]$ |
| gamma | $\{0, \text{LogUniform}[1e\text{-}3, 100]\}$ |
| lambda | $\{0, \text{LogUniform}[0.1, 10]\}$ |
| min_child_weight | LogUniform$[1e\text{-}4, 100]$ |
| subsample | Uniform$[0.5, 1.0]$ |
| colsample_bytree | Uniform$[0.5, 1.0]$ |
| early_stopping_rounds | 200 |
| # Tuning iterations | 200 |

## G. Per-Dataset Results

The per-dataset results are reported in the tables below.

*Table 14.* Per-dataset performance for the methods. For each dataset, we report the mean and standard deviation of test metric across ten random seeds.

| Churn $\uparrow$ | |
|---|---|
| Method | Score |
| MLP | $0.8562 \pm 0.0018$ |
| XGBoost | $0.8605 \pm 0.0020$ |
| TabICLv2 | $0.8658 \pm 0.0008$ |
| TabPFN$-3$ | $0.8655 \pm 0.0009$ |
| ModernNCA$^\dagger$ | $0.8591 \pm 0.0027$ |
| RealMLP | $0.8599 \pm 0.0023$ |
| TabM | $0.8638 \pm 0.0017$ |
| MLP$^\dagger$ | $0.8633 \pm 0.0029$ |
| TabM$^\dagger$ | $0.8609 \pm 0.0014$ |
| TabPack | $0.8575 \pm 0.0014$ |
| MLP$^\dagger_{\text{Ens}}$ | $0.8624 \pm 0.0013$ |
| TabPack$^\dagger_{\text{Offline}}$ | $0.8562 \pm 0.0048$ |
| TabPack$^\dagger_{\text{MPS}}$ | $0.8588 \pm 0.0052$ |
| TabPack$^\dagger$ | $0.8623 \pm 0.0028$ |

| California $\downarrow$ | |
|---|---|
| Method | Score |
| MLP | $0.4819 \pm 0.0023$ |
| XGBoost | $0.4329 \pm 0.0011$ |
| TabICLv2 | $0.3976 \pm 0.0005$ |
| TabPFN$-3$ | $0.3787 \pm 0.0007$ |
| ModernNCA$^\dagger$ | $0.4098 \pm 0.0035$ |
| RealMLP | $0.4063 \pm 0.0011$ |
| TabM | $0.4324 \pm 0.0026$ |
| MLP$^\dagger$ | $0.4376 \pm 0.0022$ |
| TabM$^\dagger$ | $0.4028 \pm 0.0019$ |
| TabPack | $0.4725 \pm 0.0014$ |
| MLP$^\dagger_{\text{Ens}}$ | $0.4233 \pm 0.0024$ |
| TabPack$^\dagger_{\text{Offline}}$ | $0.4241 \pm 0.0025$ |
| TabPack$^\dagger_{\text{MPS}}$ | $0.4171 \pm 0.0011$ |
| TabPack$^\dagger$ | $0.4176 \pm 0.0028$ |

### House ↓

| Method | Score |
|---|---|
| MLP | $30387 \pm 214$ |
| XGBoost | $31381 \pm 60$ |
| TabICLv2 | $27958 \pm 87$ |
| TabPFN$-$3 | $26913 \pm 75$ |
| ModernNCA$^\dagger$ | $30713 \pm 399$ |
| RealMLP | $30077 \pm 138$ |
| TabM | $30390 \pm 124$ |
| MLP$^\dagger$ | $29851 \pm 411$ |
| TabM$^\dagger$ | $30151 \pm 218$ |
| TabPack | $29721 \pm 91$ |
| MLP$^\dagger_{\mathrm{Ens}}$ | $29611 \pm 198$ |
| TabPack$^\dagger_{\mathrm{Offline}}$ | $29896 \pm 179$ |
| TabPack$^\dagger_{\mathrm{MPS}}$ | $29477 \pm 162$ |
| TabPack$^\dagger$ | $29549 \pm 175$ |

### Sberbank-Housing ↓

| Method | Score |
|---|---|
| MLP | $0.2608 \pm 0.0149$ |
| XGBoost | $0.2407 \pm 0.0004$ |
| TabICLv2 | $0.3137 \pm 0.0107$ |
| TabPFN$-$3 | $0.2288 \pm 0.0013$ |
| ModernNCA$^\dagger$ | $0.2379 \pm 0.0047$ |
| RealMLP | $0.2302 \pm 0.0012$ |
| TabM | $0.2424 \pm 0.0033$ |
| MLP$^\dagger$ | $0.2481 \pm 0.0038$ |
| TabM$^\dagger$ | $0.2343 \pm 0.0013$ |
| TabPack | $0.2470 \pm 0.0018$ |
| MLP$^\dagger_{\mathrm{Ens}}$ | $0.2345 \pm 0.0010$ |
| TabPack$^\dagger_{\mathrm{Offline}}$ | $0.2339 \pm 0.0023$ |
| TabPack$^\dagger_{\mathrm{MPS}}$ | $0.2309 \pm 0.0017$ |
| TabPack$^\dagger$ | $0.2302 \pm 0.0014$ |

### Adult ↑

| Method | Score |
|---|---|
| MLP | $0.8561 \pm 0.0012$ |
| XGBoost | $0.8709 \pm 0.0007$ |
| TabICLv2 | $0.8699 \pm 0.0022$ |
| TabPFN$-$3 | $0.8636 \pm 0.0002$ |
| ModernNCA$^\dagger$ | $0.8687 \pm 0.0011$ |
| RealMLP | $0.8717 \pm 0.0014$ |
| TabM | $0.8577 \pm 0.0007$ |
| MLP$^\dagger$ | $0.8692 \pm 0.0018$ |
| TabM$^\dagger$ | $0.8674 \pm 0.0010$ |
| TabPack | $0.8572 \pm 0.0011$ |
| MLP$^\dagger_{\mathrm{Ens}}$ | $0.8671 \pm 0.0012$ |
| TabPack$^\dagger_{\mathrm{Offline}}$ | $0.8680 \pm 0.0023$ |
| TabPack$^\dagger_{\mathrm{MPS}}$ | $0.8694 \pm 0.0019$ |
| TabPack$^\dagger$ | $0.8693 \pm 0.0007$ |

### Diamond ↓

| Method | Score |
|---|---|
| MLP | $0.1359 \pm 0.0012$ |
| XGBoost | $0.1337 \pm 0.0004$ |
| TabICLv2 | $0.1249 \pm 0.0002$ |
| TabPFN$-$3 | $0.1246 \pm 0.0002$ |
| ModernNCA$^\dagger$ | $0.1331 \pm 0.0018$ |
| RealMLP | $0.1308 \pm 0.0006$ |
| TabM | $0.1315 \pm 0.0010$ |
| MLP$^\dagger$ | $0.1343 \pm 0.0008$ |
| TabM$^\dagger$ | $0.1306 \pm 0.0007$ |
| TabPack | $0.1331 \pm 0.0003$ |
| MLP$^\dagger_{\mathrm{Ens}}$ | $0.1307 \pm 0.0005$ |
| TabPack$^\dagger_{\mathrm{Offline}}$ | $0.1309 \pm 0.0003$ |
| TabPack$^\dagger_{\mathrm{MPS}}$ | $0.1303 \pm 0.0003$ |
| TabPack$^\dagger$ | $0.1307 \pm 0.0007$ |

### Otto ↑

| Method | Score |
|---|---|
| MLP | $0.8218 \pm 0.0029$ |
| XGBoost | $0.8301 \pm 0.0016$ |
| TabICLv2 | $0.8434 \pm 0.0012$ |
| TabPFN$-$3 | $0.8405 \pm 0.0008$ |
| ModernNCA$^\dagger$ | $0.8298 \pm 0.0022$ |
| RealMLP | $0.8287 \pm 0.0007$ |
| TabM | $0.8278 \pm 0.0012$ |
| MLP$^\dagger$ | $0.8237 \pm 0.0023$ |
| TabM$^\dagger$ | $0.8332 \pm 0.0021$ |
| TabPack | $0.8238 \pm 0.0012$ |
| MLP$^\dagger_{\mathrm{Ens}}$ | $0.8241 \pm 0.0018$ |
| TabPack$^\dagger_{\mathrm{Offline}}$ | $0.8248 \pm 0.0020$ |
| TabPack$^\dagger_{\mathrm{MPS}}$ | $0.8292 \pm 0.0017$ |
| TabPack$^\dagger$ | $0.8281 \pm 0.0017$ |

### Higgs-Small ↑

| Method | Score |
|---|---|
| MLP | $0.7267 \pm 0.0013$ |
| XGBoost | $0.7271 \pm 0.0009$ |
| TabICLv2 | $0.7355 \pm 0.0009$ |
| TabPFN$-$3 | $0.7391 \pm 0.0008$ |
| ModernNCA$^\dagger$ | $0.7310 \pm 0.0014$ |
| RealMLP | $0.7311 \pm 0.0014$ |
| TabM | $0.7409 \pm 0.0029$ |
| MLP$^\dagger$ | $0.7279 \pm 0.0007$ |
| TabM$^\dagger$ | $0.7352 \pm 0.0008$ |
| TabPack | $0.7288 \pm 0.0021$ |
| MLP$^\dagger_{\mathrm{Ens}}$ | $0.7313 \pm 0.0008$ |
| TabPack$^\dagger_{\mathrm{Offline}}$ | $0.7309 \pm 0.0007$ |
| TabPack$^\dagger_{\mathrm{MPS}}$ | $0.7316 \pm 0.0018$ |
| TabPack$^\dagger$ | $0.7317 \pm 0.0010$ |

### Black-Friday ↓

| Method | Score |
| --- | --- |
| MLP | $0.6927 \pm 0.0006$ |
| XGBoost | $0.6808 \pm 0.0001$ |
| TabICLv2 | $0.6919 \pm 0.0002$ |
| TabPFN−3 | $0.7026 \pm 0.0028$ |
| ModernNCA$^\dagger$ | $0.6861 \pm 0.0006$ |
| RealMLP | $0.6781 \pm 0.0004$ |
| TabM | $0.6846 \pm 0.0003$ |
| MLP$^\dagger$ | $0.6816 \pm 0.0006$ |
| TabM$^\dagger$ | $0.6766 \pm 0.0006$ |
| TabPack | $0.6877 \pm 0.0003$ |
| MLP$^\dagger_{\text{Ens}}$ | $0.6802 \pm 0.0003$ |
| TabPack$^\dagger_{\text{Offline}}$ | $0.6797 \pm 0.0005$ |
| TabPack$^\dagger_{\text{MPS}}$ | $0.6783 \pm 0.0001$ |
| TabPack$^\dagger$ | $0.6784 \pm 0.0002$ |

### Ecom-Offers ↑

| Method | Score |
| --- | --- |
| MLP | $0.6011 \pm 0.0025$ |
| XGBoost | $0.5741 \pm 0.0055$ |
| TabICLv2 | $0.5974 \pm 0.0020$ |
| TabPFN−3 | $0.6343 \pm 0.0009$ |
| ModernNCA$^\dagger$ | $0.5831 \pm 0.0019$ |
| RealMLP | $0.5806 \pm 0.0145$ |
| TabM | $0.6016 \pm 0.0004$ |
| MLP$^\dagger$ | $0.5993 \pm 0.0008$ |
| TabM$^\dagger$ | $0.5984 \pm 0.0010$ |
| TabPack | $0.6018 \pm 0.0012$ |
| MLP$^\dagger_{\text{Ens}}$ | $0.5986 \pm 0.0017$ |
| TabPack$^\dagger_{\text{Offline}}$ | $0.5987 \pm 0.0009$ |
| TabPack$^\dagger_{\text{MPS}}$ | $0.5990 \pm 0.0013$ |
| TabPack$^\dagger$ | $0.5989 \pm 0.0017$ |

### Maps-Routing ↓

| Method | Score |
| --- | --- |
| MLP | $0.1622 \pm 0.0001$ |
| XGBoost | $0.1618 \pm 0.0000$ |
| TabICLv2 | $0.1657 \pm 0.0002$ |
| TabPFN−3 | $0.1643 \pm 0.0002$ |
| ModernNCA$^\dagger$ | $0.1628 \pm 0.0000$ |
| RealMLP | $0.1609 \pm 0.0001$ |
| TabM | $0.1611 \pm 0.0001$ |
| MLP$^\dagger$ | $0.1612 \pm 0.0001$ |
| TabM$^\dagger$ | $0.1606 \pm 0.0002$ |
| TabPack | $0.1612 \pm 0.0000$ |
| MLP$^\dagger_{\text{Ens}}$ | $0.1607 \pm 0.0001$ |
| TabPack$^\dagger_{\text{Offline}}$ | $0.1610 \pm 0.0001$ |
| TabPack$^\dagger_{\text{MPS}}$ | $0.1604 \pm 0.0001$ |
| TabPack$^\dagger$ | $0.1605 \pm 0.0001$ |

### Homesite-Insurance ↑

| Method | Score |
| --- | --- |
| MLP | $0.9515 \pm 0.0007$ |
| XGBoost | $0.9606 \pm 0.0001$ |
| TabICLv2 | $0.9470 \pm 0.0046$ |
| TabPFN−3 | $0.9568 \pm 0.0015$ |
| ModernNCA$^\dagger$ | $0.9629 \pm 0.0003$ |
| RealMLP | $0.9600 \pm 0.0015$ |
| TabM | $0.9652 \pm 0.0004$ |
| MLP$^\dagger$ | $0.9627 \pm 0.0007$ |
| TabM$^\dagger$ | $0.9644 \pm 0.0007$ |
| TabPack | $0.9518 \pm 0.0002$ |
| MLP$^\dagger_{\text{Ens}}$ | $0.9638 \pm 0.0004$ |
| TabPack$^\dagger_{\text{Offline}}$ | $0.9640 \pm 0.0004$ |
| TabPack$^\dagger_{\text{MPS}}$ | $0.9642 \pm 0.0001$ |
| TabPack$^\dagger$ | $0.9644 \pm 0.0003$ |

### Cooking-Time ↓

| Method | Score |
| --- | --- |
| MLP | $0.4824 \pm 0.0002$ |
| XGBoost | $0.4824 \pm 0.0001$ |
| TabICLv2 | $0.4851 \pm 0.0004$ |
| TabPFN−3 | $0.4844 \pm 0.0004$ |
| ModernNCA$^\dagger$ | $0.4813 \pm 0.0003$ |
| RealMLP | $0.4809 \pm 0.0005$ |
| TabM | $0.4807 \pm 0.0002$ |
| MLP$^\dagger$ | $0.4810 \pm 0.0001$ |
| TabM$^\dagger$ | $0.4802 \pm 0.0002$ |
| TabPack | $0.4810 \pm 0.0000$ |
| MLP$^\dagger_{\text{Ens}}$ | $0.4795 \pm 0.0003$ |
| TabPack$^\dagger_{\text{Offline}}$ | $0.4798 \pm 0.0002$ |
| TabPack$^\dagger_{\text{MPS}}$ | $0.4793 \pm 0.0002$ |
| TabPack$^\dagger$ | $0.4792 \pm 0.0002$ |

### Homecredit-Default ↑

| Method | Score |
| --- | --- |
| MLP | $0.8551 \pm 0.0009$ |
| XGBoost | $0.8675 \pm 0.0003$ |
| TabICLv2 | $0.8454 \pm 0.0019$ |
| TabPFN−3 | $0.8501 \pm 0.0014$ |
| ModernNCA$^\dagger$ | $0.8526 \pm 0.0019$ |
| RealMLP | $0.8509 \pm 0.0023$ |
| TabM | $0.8613 \pm 0.0011$ |
| MLP$^\dagger$ | $0.8601 \pm 0.0004$ |
| TabM$^\dagger$ | $0.8642 \pm 0.0006$ |
| TabPack | $0.8535 \pm 0.0003$ |
| MLP$^\dagger_{\text{Ens}}$ | $0.8615 \pm 0.0015$ |
| TabPack$^\dagger_{\text{Offline}}$ | $0.8617 \pm 0.0010$ |
| TabPack$^\dagger_{\text{MPS}}$ | $0.8611 \pm 0.0007$ |
| TabPack$^\dagger$ | $0.8615 \pm 0.0006$ |

| Delivery-Eta ↓ | |
|---|---|
| Method | Score |
| MLP | $0.5476 \pm 0.0005$ |
| XGBoost | $0.5458 \pm 0.0001$ |
| TabICLv2 | $0.5537 \pm 0.0003$ |
| TabPFN−3 | $0.5545 \pm 0.0009$ |
| ModernNCA$^\dagger$ | $0.5508 \pm 0.0003$ |
| RealMLP | $0.5468 \pm 0.0009$ |
| TabM | $0.5483 \pm 0.0002$ |
| MLP$^\dagger$ | $0.5499 \pm 0.0008$ |
| TabM$^\dagger$ | $0.5464 \pm 0.0004$ |
| TabPack | $0.5502 \pm 0.0004$ |
| MLP$^\dagger_\text{Ens}$ | $0.5490 \pm 0.0009$ |
| TabPack$^\dagger_\text{Offline}$ | $0.5486 \pm 0.0009$ |
| TabPack$^\dagger_\text{MPS}$ | $0.5474 \pm 0.0007$ |
| TabPack$^\dagger$ | $0.5477 \pm 0.0005$ |

| Weather ↓ | |
|---|---|
| Method | Score |
| MLP | $1.5010 \pm 0.0037$ |
| XGBoost | $1.4694 \pm 0.0005$ |
| TabICLv2 | $1.4690 \pm 0.0029$ |
| TabPFN−3 | $1.4597 \pm 0.0011$ |
| ModernNCA$^\dagger$ | $1.5007 \pm 0.0028$ |
| RealMLP | $1.4398 \pm 0.0016$ |
| TabM | $1.4651 \pm 0.0042$ |
| MLP$^\dagger$ | $1.4915 \pm 0.0020$ |
| TabM$^\dagger$ | $1.4491 \pm 0.0034$ |
| TabPack | $1.4742 \pm 0.0010$ |
| MLP$^\dagger_\text{Ens}$ | $1.4607 \pm 0.0021$ |
| TabPack$^\dagger_\text{Offline}$ | $1.4659 \pm 0.0040$ |
| TabPack$^\dagger_\text{MPS}$ | $1.4436 \pm 0.0013$ |
| TabPack$^\dagger$ | $1.4430 \pm 0.0007$ |

| Microsoft ↓ | |
|---|---|
| Method | Score |
| MLP | $0.7455 \pm 0.0003$ |
| XGBoost | $0.7412 \pm 0.0001$ |
| TabICLv2 | $0.7457 \pm 0.0002$ |
| TabPFN−3 | $0.7785 \pm 0.0026$ |
| ModernNCA$^\dagger$ | $0.7443 \pm 0.0005$ |
| RealMLP | $0.7446 \pm 0.0009$ |
| TabM | $0.7422 \pm 0.0004$ |
| MLP$^\dagger$ | $0.7447 \pm 0.0002$ |
| TabM$^\dagger$ | $0.7407 \pm 0.0002$ |
| TabPack | $0.7423 \pm 0.0002$ |
| MLP$^\dagger_\text{Ens}$ | $0.7421 \pm 0.0005$ |
| TabPack$^\dagger_\text{Offline}$ | $0.7424 \pm 0.0002$ |
| TabPack$^\dagger_\text{MPS}$ | $0.7395 \pm 0.0002$ |
| TabPack$^\dagger$ | $0.7395 \pm 0.0002$ |

