# OpenReview forum: "TabPack: Efficient Hyperparameter Ensembles for Tabular Deep Learning"
_ICML.cc/2026/Conference — ICML 2026 regular_

### Official Review · Reviewer_5wYi · 2026-03-07

**Soundness:** 2
**Presentation:** 3
**Significance:** 1
**Originality:** 2
**Overall Recommendation:** 4
**Confidence:** 2

**Summary:**

This paper proposes TabPack, an ensemble-first framework for tabular deep learning. The main idea is to pack many MLP-like base models with different model and optimizer hyperparameters into a single packed model-optimizer pair, train them in parallel in one run, and build the final ensemble online during training.

**Compliance With Llm Reviewing Policy:**

Affirmed.

**Final Justification:**

My main concerns have been addressed by the rebuttal. So I raise my rating to weak accept. I suggest the author make a comparison with AutoGluon.

**Key Questions For Authors:**

See weaknesses.

**Limitations:**

Yes

**Strengths And Weaknesses:**

### Strengths
- I found the method easy to understand, and the paper does a good job explaining the main components.
- In tabular deep learning, ensembles often help a lot in practice, but the usual workflow of tuning and training many models independently is slow and cumbersome. This paper takes that issue seriously and proposes an ensemble-first design instead of treating ensembling as an afterthought.
- The packed implementation is natural for modern hardware, and the reported runtime gains over standard tuning pipelines are large.

### Weaknesses
- The runtime comparison is somewhat hard to interpret as a pure model comparison. For the baselines, runtime is the full hyperparameter tuning time, while for TabPack and MLPEns it is the time of a single run with no tuning. I understand that this is part of the intended use case, and in that sense the result is meaningful. Still, the comparison mixes two things: model quality and workflow design. So the paper clearly shows that TabPack is a very efficient experiment pipeline, but it is a bit less clear how much of the gain should be interpreted as a strictly better learning method versus a different evaluation protocol. I would have liked this distinction to be stated more explicitly.
- A second concern is that the evidence for the online-ensemble advantage is not yet fully isolated. The paper concludes that online ensembling is better than offline ensembling based on comparisons to TabPackOffline and MLPEns. However, MLPEns differs from TabPack in more than just online versus offline ensembling, since it is also implemented as a multi-process isolated-training pipeline rather than a packed joint system. So this comparison combines several effects at once. The TabPackOffline comparison is more relevant, but I would still have liked a cleaner ablation focused directly on the contribution of online ensemble updates and ensemble-based early stopping.
- Another weakness is the baseline coverage. While the paper includes strong GBDT baselines, plain MLP, ModernNCA, and TabM, it does not compare against some important recent tabular baselines, especially stronger tuned MLP-style baselines such as RealMLP, and representative recent tabular foundation-model methods such as TabPFN, TabICL, and LimiX. Since TabPack is positioned as a practical go-to solution for tabular deep learning, I think broader coverage of strong recent baselines would make the empirical claims more convincing.

If the author can address my questions, I will consider raising my score.

---

> ### Author Rebuttal · Authors · 2026-03-31
>
> We thank the reviewer for the comments, and we truly appreciate the attention to detail. We are looking forward to the discussion.
>
> ### Positive things
>
> One thing we are especially excited to see in the review is that we are on the same page regarding the mismatch of how useful ensembles can be and the current challenges of training them. And, as the review says, _"This paper takes that issue seriously and proposes an ensemble-first design instead of treating ensembling as an afterthought."_ This is indeed the key technical motivation behind our method.
>
> ### Other topics
>
> > The runtime comparison is somewhat hard to interpret ... the result is meaningful ... the paper clearly shows that TabPack is a very efficient experiment pipeline ... the comparison mixes two things ... I would have liked this distinction to be stated more explicitly.
>
> _TL;DR_
>
> **The paper already allows reasoning about _all_ of the following:**
>
> - Performance and efficiency aspects _in isolation_.
> - Performance and efficiency aspects _in combination_.
> - What exactly makes TabPack runtime efficient.
>
> _Details_
>
> We should honestly say that it is a bit unclear what the concern is exactly, especially given that it mixes reservations and somewhat positive things. We will do our best to sort out all the important points, and we are open to further discussion and clarification.
>
> _First_, the paper already provides some _independent_ analysis of performance- and efficiency-related aspects. Examples:
>
> - Efficiency: Figure 2, Figure 5 (middle, right)
> - Performance: Figure 5 (left), Figure 7
>
> _Second_, the paper already provides some _joint_ analysis of performance- and efficiency-related aspects. Example: Figure 6.
>
> _Third_, the overall runtime efficiency of TabPack originates from two things:
>
> - _It runs only once._ This is true thanks to the _ML algorithm_ itself (parallel training and online ensembling of many models with randomly sampled hyperparameters). The fact that running TabPack only once is enough is already illustrated by Figure 5 (left).
> - _It utilizes GPU well_. This is where the _packed implementation_ comes into play, as already shown in Figure 2.
>
> If there is anything else we can clarify, please let us know.
>
> > A second concern ....
>
> _TL;DR_
>
> - **The second concern is missing one technical detail, which automatically resolves the concern**.
> - The paper already allows for a clean comparison of online and offline ensembling.
>
> _Details_
>
> The reviewer seems to suggest that TabPack-Offline somehow differs from the MLP-Ens baseline on the _algorithmic_ level (please correct us if we are missing anything). **This is not the case.** To resolve the confusion, we summarize the key points below.
>
> _First_, as an ML algorithm, MLP-Ens consists of two steps:
>
> - Train $m$ based models with randomly sampled hyperparameters in isolation _without_ ensemble-based early stopping.
> - Build an ensemble (greedily) out of the $m$ trained base models.
>
> _Second_, **TabPack-Offline is exactly the same ML algorithm as MLP-Ens,** and differs from it only in its technical implementation. In particular:
>
> - Similarly to MLP-Ens, _there is no ensemble-based early stopping in TabPack-Offline._
> - The numerical difference in the performance between TabPack-Offline and MLP-Ens _fully_ comes from the difference in their technical implementations (packed vs multiprocessing-based), which implies the difference in all usual sources of randomness (initialization, training batches, dropout masks, etc.) and computation kernels (e.g., matmul in MLP-Ens and batched matmul in TabPack-Offline).
>
> _Third,_ the high-level difference of TabPack from TabPack-Offline is:
>
> - The online nature, which implies using _intermediate_ base model checkpoints for building the ensemble.
> - The ensemble-based early stopping.
>
> _Lastly_, since the ensemble-based early stopping cannot _improve_ the validation performance (since it truncates the hypothetical full training history), comparing TabPack against TabPack-Offline and/or MLP-Ens is enough for reasoning about the difference between online and offline ensembling.
>
> > Another weakness is the baseline coverage.
>
> _TL;DR_
>
> **TabPack outperforms new baselines, and our claims hold.**
>
> _Details_
>
> _First_, the results with TabICL-V2 (the latest state-of-the-art TFM), RealMLP, and baselines suggested by other reviewers are available here:
>
> https://freeimage.host/i/B3CgTGf
>
> _Second_, the suggested TabPFN and LimiX do not scale (OOM) to many of our datasets, and thus are not included.
>
> _Third_, as a bonus point, we note that despite the good task performance of TabPack on our benchmark, we don't build our story solely around that. To learn more about the vision of our project, we invite the reviewer to read the "Big Picture" part of our reply to Reviewer `91be`.
>
> ---
>
> We thank the reviewer again for the detailed feedback, and we will be happy to continue the discussion.

---

> > ### Author Rebuttal · Reviewer_5wYi · 2026-04-01
> >
> > I thank the authors for addressing my concerns. I have updated my score.

---

### Official Review · Reviewer_QYgx · 2026-03-09

**Soundness:** 3
**Presentation:** 2
**Significance:** 2
**Originality:** 2
**Overall Recommendation:** 3
**Confidence:** 5

**Summary:**

This paper proposes TabPack, an efficient hyperparameter ensemble method for deep learning on tabular data. This method packages multiple base models and optimizer pairs with different hyperparameters into a single neural network. Through parallel training and online ensemble selection, it builds a powerful ensemble model in a single run, significantly reducing the need for traditional hyperparameter tuning. Experiments on 17 public benchmark datasets demonstrate that TabPack achieves substantially higher efficiency while maintaining performance comparable to state-of-the-art methods.

**Compliance With Llm Reviewing Policy:**

Affirmed.

**Final Justification:**

Thanks to the author for responding to all my concerns and follow-up questions, I would like to hold my assessment.

**Key Questions For Authors:**

# Strengths
S1: TabPack completes ensemble training in a single run, reducing training time from tens of hours (e.g., 47.2 hours for TabM) to just a few hours (2.4 hours), achieving a substantial efficiency gain and providing practitioners with a tool for quickly evaluating model potential.

S2: By randomly sampling hyperparameters, TabPack reduces reliance on traditional hyperparameter tuning, lowering the barrier to entry and enabling researchers to quickly test the performance of new architectures within an ensemble framework.

S3: This paper demonstrates that online ensemble methods (dynamically selecting ensemble members during training) outperform offline methods. This is a valuable finding that provides direction for future research.

# Weaknesses
W1: The core idea of ​​TabPack is highly similar to TabM and Packed Ensembles, with the main difference being the hyperparameter sampling strategy.

W2: It does not provide sufficient comparison with state-of-the-art tabular models (such as TabPFN, FT-Transformer) or powerful tree models (LightGBM), and most of the 17 datasets are small to medium-sized, with some results showing only minor differences but without statistical significance testing.

W3: The paper acknowledges that TabPack requires more GPU memory for training but does not provide a detailed analysis of memory consumption. For resource-constrained scenarios, this limitation could severely impact its practicality.

**Limitations:**

Q1: The paper mentions that TabPack requires more GPU memory. Could you provide a specific analysis of its memory consumption? When the number of base models (m) increases or the dataset size expands, will the packed training method encounter a memory bottleneck, thus limiting its application in large-scale industrial scenarios?

Q2: TabPack currently primarily achieves efficient parallel hyperparameter ensembles for MLP architectures. Is this methodology generalizable and can it be extended to Transformer-based tabular deep learning models? Furthermore, considering that TabPack is essentially an explicit ensemble achieved by increasing "width," while the Transformer architecture tends to achieve "deep" ensembles through hierarchical structures, I believe a thorough comparison of the representational capabilities and generalization performance of these two different ensemble paradigms on tabular data is important.

**Strengths And Weaknesses:**

Please provide a thorough assessment of the strengths and weaknesses of the paper, touching on each of the following dimensions: soundness, presentation, significance, and originality. We encourage you to be open-minded about the potential strengths and broad definitions of significance and originality. For example, originality may arise from creative combinations of existing ideas, application to a real-world use case, or removing restrictive assumptions from prior theoretical results. We provide detailed guidelines below on each dimension:

---

> ### Author Rebuttal · Authors · 2026-03-31
>
> We thank the reviewer for the comments, and we are excited about the upcoming discussion. We truly hope to find a common ground on the suggested topics.
>
> ### Positive things
>
> Among the mentioned strengths, we are especially glad to see S1 (_"**providing practitioners with a tool for quickly evaluating** model potential"_) and S2 (_"TabPack reduces reliance on traditional hyperparameter tuning ... **enabling researchers to quickly test** the performance of new architectures"_). Indeed, **empowering practitioners and researchers** with a training-efficient tuning-free workflow delivering competitive performance is a major goal of our work.
>
> ### Big picture
>
> We highlight the following points as an important context for the discussion:
>
> - _**The main goal of our paper**: in the niche of **non-foundation tabular DL models**, unlock **faster experiment cycles** with **less hyperparameter tuning**, while maintaining **competitive task performance**._ In particular, we are _equally_ focused on _all_ parts of Figure 5 and Figure 6 of the paper.
> - _**The main result**: our method achieves competitive performance in a single run without hyperparameter tuning_. This is a **major shift** and a **new workflow** for non-foundation tabular DL models.
> - _**The main purpose of our experiments** is to prove the feasibility, practicality, and competitiveness of the new workflow, not to become the new SOTA on as many datasets as possible_.
>
> ### Other topics
>
> > It does not provide sufficient comparison with state-of-the-art tabular models (such as TabPFN, FT-Transformer) or powerful tree models (LightGBM)
>
> Good baseline coverage is important to us, however, **the suggested baselines cannot affect our story.** Specifically:
>
> - **TabPFN is not applicable**, since it does not scale (i.e., OOMs) to many of our datasets.
> - **FT-Transformer has already been outperformed** in `[4]`, and we use the experiment protocol and datasets from `[4]`.
> - **Our paper already includes XGBoost**, and LightGBM has already been shown to perform similarly to it in `[4]`, and we use the experiment protocol and datasets from `[4]`.
>
> **Nevertheless, we provide new results** with TabICL V2 `[5]` (the latest SOTA tabular foundation model), FT-Transformer, LightGBM, and RealMLP (suggested by another reviewer) here:
>
> https://freeimage.host/i/B3CgTGf
>
> Summary: **TabPack outperforms all the new baselines.**
>
> > datasets are small to medium-sized
>
> To illustrate that **TabPack is applicable to large datasets**, we  report the mean and standard deviations of RMSE on two **large-scale industrial datasets** from the TabReD benchmark `[1]` (model names follow Figure 5):
>
> | Dataset | #Objects | #Features | $\mathrm{XGBoost}$ | $\mathrm{TabM}^\dagger$ | $\mathrm{TabPack}^\dagger$ |
> | --- | --- | --- | --- | --- | --- |
> | Maps Routing | 6.5M | 986 | 0.1601 $\pm$ 0.0001 | 0.1595 $\pm$ 0.0009 | **0.1572  $\pm$ 0.0001** |
> | Weather | 13M | 13 | 1.4234  $\pm$ 0.0010 | 1.4065  $\pm$ 0.0052 | **1.3890  $\pm$  0.0004** |
>
> We highlight the following:
>
> - Above, **TabPack achieves the best results**. And again, **in a single run without hyperparameter tuning.**
> - In many recent tabular DL papers, the maximum dataset size is less than 1M `[2]`, `[3]`, `[5]`.
>
> > statistical significance
>
> Please note:
>
> - The score used to sort methods in Figure 5 (left) does _not_ depend on pairwise comparisons, so it is unclear what statistical testing can be applied here.
> - **The ranks reported in Figure 5 already take standard deviations over random seeds into account.**
>
> As a reminder, as mentioned in L599 of our paper, we inherit the experiment protocol (including the relative-improvement-over-MLP score and the rank definition) from `[4]`.
>
> > memory consumption
>
> Below, we report the mean and standard deviation of the memory usage during training (as reported by the `reserved_bytes.all.peak` output of `torch.cuda.memory.memory_stats`) across all datasets:
>
> - TabM: $3.8 \pm 5.2$ GB
> - TabPack: $10.6 \pm 14.1$ GB
>
>
> > When the number of base models (m) increases or the dataset size expands, will the packed training method encounter a memory bottleneck, thus limiting its application in large-scale industrial scenarios?
>
> As shown earlier, **TabPack successfully runs on large-scale industrial datasets.** In particular, there was no need to increase $m$ for achieving good performance.
>
> ### References
>
> - `[1]` (ICLR 2025) TabReD: Analyzing Pitfalls and Filling the Gaps in Tabular Deep Learning Benchmarks
> - `[2]` (ICLR 2025) Revisiting Nearest Neighbor for Tabular Data: A Deep Tabular Baseline Two Decades Later
> - `[3]` (NeurIPS 2024) Better by Default: Strong Pre-Tuned MLPs and Boosted Trees on Tabular Data
> - `[4]` (ICLR 2025) TabM: Advancing Tabular Deep Learning with Parameter-Efficient Ensembling
> - `[5]` (2026) TabICLv2: A better, faster, scalable, and open tabular foundation model

---

> > ### Author Rebuttal · Reviewer_QYgx · 2026-04-02
> >
> > We appreciate the response and valid experiments resolving most concerns, though questions remain.
> >
> > Q1: Given both methods focus on efficient tabular ensembling and the reliance on TabM's protocol, please clarify TabPack's novel methodological contributions compared to TabM's parameter-efficient ensembling mechanism.
> >
> > Q2: TabPack's reported 10.6GB mean memory versus TabM's 3.8GB is substantial. The Weather dataset's low dimensionality of 13 features bypasses typical memory pressures of wide tabular inputs. Please conduct a progressive scalability stress-test detailing memory footprint growth when scaling ensemble size on a moderately wide dataset, and discuss this scaling behavior in the Limitations section.

---

> > > ### Author Response · Authors · 2026-04-03
> > >
> > > _NOTE: According to ICML rules, this is our final post in the thread. We thank the reviewer for the discussion!_
> > >
> > > We thank the reviewer for the additional questions. We answer them below.
> > >
> > > > please clarify TabPack's novel methodological contributions compared to TabM's parameter-efficient ensembling mechanism
> > >
> > > _First_, we show that TabM and TabPack are two different methods, with the only similarity being that both represent MLP ensembles:
> > >
> > > | | TabM | TabPack |
> > > | --- | --- | --- |
> > > | Model type | MLP Ensemble | MLP Ensemble |
> > > | Ensemble method | BatchEnsemble or MiniEnsemble | **Packed Hyperparameter Ensemble** |
> > > | Ensemble member selection | No selection | **Greedy** |
> > > | Base method hyperparameters | Same for all members | **Randomly sampled** |
> > > | Hyperparameter tuning | Required for top performance | **Not required** |
> > >
> > > _Second_, the key technical contribution of our work is the packed approach to hyperparameter ensembling. The detailed comparison with prior work is available in the Related Work section of our paper. In short:
> > >
> > > - We diversify base models by as many degrees as possible (depths, widths, dropout rates, feature embedding sizes, etc.), while prior work explored this opportunity only to a limited extent.
> > > - Moreover, we also diversify _optimizers_ (learning rates, weight decays, etc.). This makes TabPack even more unique compared to prior work.
> > > - (note: TabM does _none_ of the above; instead, TabM focuses on the parameter efficiency)
> > >
> > > > The Weather dataset's low dimensionality of 13 features
> > >
> > > This is a typo in our rebuttal: **the actual number of features is 103, not 13**. We thank the reviewer for pointing to this. As a quick summary, in the rebuttal, we covered two large-scale datasets:
> > >
> > > - Maps Routing: **6.5M objects and 986 features** (large and wide)
> > > - Weather: **13M objects and 103 features** (even larger, moderately wide)
> > >
> > > Note that the "Maps Routing" dataset with **986 features** is already a serious test for models.
> > >
> > > > Please conduct a progressive scalability stress-test detailing memory footprint growth when scaling ensemble size on a moderately wide dataset
> > >
> > > The following table reports the peak GPU memory usage during training for TabM and TabPack on the Weather dataset (103 features):
> > >
> > > | Ensemble size | TabM GPU RAM (GB) | TabPack GPU RAM (GB) |
> > > | ---: | ---: | ---: |
> > > | 1 | 0.31 | 0.35 |
> > > | 2 | 0.36 | 0.48 |
> > > | 4 | 0.59 | 0.78 |
> > > | 8 | 0.97 | 1.26 |
> > > | 16 | 1.51 | 2.21 |
> > > | 32 | 3.06 | 3.97 |
> > > | 64 | 5.93 | 7.74 |
> > > | 128 | 11.15 | 14.75 |
> > >
> > > In the above table, for both models, the base MLP depth is set to 3, and the base width is set to 384 (these are the official default values for both TabM and TabPack).
> > >
> > > > TabPack's reported 10.6GB mean memory versus TabM's 3.8GB is substantial
> > >
> > > The message behind these numbers is that **"both TabM and TabPack can be easily trained on one modern GPU"**. For example, on the same hardware (A100 80GB), this is not the case for the previously suggested baselines:
> > >
> > > - **FT-Transformer OOMs on the Maps Routing dataset** without additional measures.
> > > - **TabPFN OOMs on multiple datasets** of our benchmark, even without considering additional large-scale tasks.
> > >
> > > ---
> > >
> > > **Bonus part: training TabPack on a MacBook**
> > >
> > > We share the reviewer's enthusiasm about efficiency-related topics. Perhaps, the reviewer will be interested in the following fun fact: the main TabPack experiments (i.e. $\mathrm{TabPack}^\dagger$ from Figure 5) can be conducted on a recent MacBook Pro in a reasonable time without using a discrete GPU at all.
> > >
> > > As a quick illustration, the following table shows runtimes from the middle part of Figure 5 on the largest dataset used in the paper (**723K objects and 136 features**) for TabPack and TabM:
> > >
> > > | Model                      | Device                      | Runtime |
> > > | :------------------------- | :-------------------------- | ------: |
> > > | $\mathrm{TabM}^\dagger$    | NVIDIA A100                 | 12h 17m |
> > > | $\mathrm{TabPack}^\dagger$ | Apple M4 Pro GPU (20 Cores) |  3h 46m |
> > > | $\mathrm{TabPack}^\dagger$ | NVIDIA A100                 |  0h 24m |
> > >
> > > Thus, TabPack makes modern tabular DL not only faster and easier to use due to its single-run workflow, but also more accessible in terms of hardware requirements. We believe that this is an exciting practical outcome of our work.
> > >
> > > P.S. Perhaps, we should add the TabPack-on-MacBook experiments to Figure 5 in the camera-ready version.

---

### Official Review · Reviewer_91be · 2026-03-13

**Soundness:** 3
**Presentation:** 4
**Significance:** 4
**Originality:** 3
**Overall Recommendation:** 4
**Confidence:** 4

**Summary:**

This paper proposes TabPack, a system that trains MLP-like models with the explicit goal of using them in an ensemble for tabular prediction tasks. TabPack efficiently trains a population of candidate models by stacking their parameters along a new “pack” dimension, enabling parallel training. The system also implicitly tunes the models’ hyperparameters by initializing the model population with randomly sampled hyperparameters and choosing ensemble members through greedy selection.
The proposed system is empirically evaluated on tabular benchmarks and achieves similar performance to comparable methods while being multiple times more efficient.

**Compliance With Llm Reviewing Policy:**

Affirmed.

**Final Justification:**

The authors addressed several of my concerns and provided the results of the optimizer ablation, making their design decisions more transparent.

The rebuttal clarifies the paper's intended scope, namely the runtime-aware performance of non-foundation tabular deep learning models on medium-to-large datasets. This restricts the empirical evaluation to a specific regime and does not fully establish the method's limits when compared to broader classes of modern approaches. In particular, while the rebuttal includes comparisons to tabular foundation models, these models are evaluated in settings in which they are known not to perform best (e.g., large-scale datasets).

Overall, the rebuttal improves the paper and increases my confidence in its contributions. However, I still find that the evaluation does not adequately probe the method’s limitations, and therefore I do not change my original score of 4.

**Key Questions For Authors:**

**Q1:** Tabular Foundation models, such as TabPFNv2.5 (Grinsztajn et al., 2026) and TabICLv2 (Qu et al., 2026), are state-of-the-art for tabular prediction. How does TabPack perform compared to such models? An evaluation of TabPack on TabArena (Erickson et al., 2025) would create more clarity.

**Q2:** The paper states in the "Optimizer family" paragraph that Muon outperforms other optimizers.  How large is the difference between different optimizers? Could the authors report these results?

**Q3:** The paper reported that TabM's hyperparameter were kept fixed. Do the authors hold any insights into TabM’s sensitivity to changes in its hyperparameter?

**Limitations:**

yes

**Strengths And Weaknesses:**

**Soundness**

The paper's main claims about TabPack’s performance and efficacy are empirically well supported. The evaluation includes strong baselines whose performance is matched by TabPack while having substantially lower run time

However, the authors refer to experiments in the "Optimizer family" paragraph to justify the use of Muon, but these experiments are not presented. Including a table in the appendix showing the optimizer's effect would improve transparency and strengthen the support for this design decision.

**Presentation**

The paper is clearly written and well structured. The descriptions of both the evaluation setup and the TabPack system are clear. The method is well motivated by placing it in the context of related work such as TabM (Gorishniy et al., 2025) and Packed Ensembles (Laurent et al., 2023). In addition, Figure 1 provides a helpful high-level overview of the training and packing procedure.

**Significance**

This paper proposes a system that trains MLP-like models with the explicit goal of using them in an ensemble for tabular prediction tasks. Improving methods for predictive tabular machine learning is an important problem with many applications across several domains, such as marketing, finance, and healthcare (Erickson et al., 2025). The proposed approach provides an efficient way to construct ensembles and may serve as a basis for further research on online ensembling methods for tabular models.

**Originality**

TabPack builds on ideas from TabM (Gorishniy et al., 2025) and Packed Ensembles (Laurent et al., 2023). However, it differs from these methods by introducing a greedy online ensembling procedure and by constructing a population of models with diverse hyperparameters, including optimizer settings. In addition, the paper provides the empirical insight that hyperparameter diversity is not necessary to achieve strong ensemble performance. Taken together, these aspects make the paper reasonably novel.

---

> ### Author Rebuttal · Authors · 2026-03-31
>
> We thank the reviewer for the detailed comments, and we are happy to share additional results and answer the questions.
>
> ### Positive things
>
> We are truly glad to read good things about the soundness, presentation, significance, and originality of our work. We are especially excited that our work is seen as a potential **"basis for further research"**. Indeed, we believe that TabPack offers a fresh approach to supervised ML tasks on tabular data, and can become one of the go-to frameworks for practitioners and researchers.
>
> ### Big picture
>
> We highlight the following points as an important context for the discussion:
>
> - _**The main goal of our paper**: in the niche of **non-foundation tabular DL models**, unlock **faster experiment cycles** with **less hyperparameter tuning**, while maintaining **competitive task performance**._ In particular, we are _equally_ focused on _all_ parts of Figure 5 and Figure 6 of the paper.
> - _**The main result**: our method achieves competitive performance in a single run without hyperparameter tuning_. This is a **major shift** and a **new workflow** for non-foundation tabular DL models.
> - _**The main purpose of our experiments** is to prove the feasibility, practicality, and competitiveness of the new workflow, not to become the new SOTA on as many datasets as possible_.
>
> ### Other topics
>
> > How large is the difference between different optimizers? Could the authors report these results?
>
> The following table reports mean and standard deviations of the relative improvement over MLP-AdamW across all datasets for different combinations of models (the notation follows Figure 5) and optimizers:
>
> | | AdamW | Muon |
> | --- | --- | --- |
> | $\mathrm{MLP}^\dagger$ | 1.1% $\pm$ 2.2% | 1.4% $\pm$ 1.4% |
> | $\mathrm{TabM}^\dagger$ | 2.3% $\pm$ 3.7% | 2.4% $\pm$ 3.1% |
> | $\mathrm{TabPack}^\dagger$ | 2.5% $\pm$ 3.7% | 2.8% $\pm$ 3.7% |
>
> The results show that **our choice of Muon is beneficial for both TabPack and for the baselines**. Importantly, all AdamW-based experiments are thoroughly tuned from scratch and evaluated under exactly the same protocol as the Muon-based experiments.
>
> > Tabular Foundation models, such as TabPFNv2.5 (Grinsztajn et al., 2026) and TabICLv2 (Qu et al., 2026), are state-of-the-art for tabular prediction. How does TabPack perform compared to such models?
>
> The results with more baselines, including TabICL V2 (the latest state-of-the-art Tabular Foundation Model), are available here:
>
> https://freeimage.host/i/B3CgTGf
>
> As one can see, **TabPack performs noticeably better than the Tabular Foundation Model (TFM) on our benchmark**. Note that _this is totally expected_, because our scope is middle-to-large datasets and includes real-world industrial datasets (the TabReD benchmark and the Microsoft dataset). To the best of our knowledge, as of now, modern foundation models struggle in such setups. **As for TabPFN 2.5, it does not scale (OOM) to many of our datasets.**
>
> > An evaluation of TabPack on TabArena (Erickson et al., 2025) would create more clarity.
>
> TabArena is tricky to fit into our story for two reasons:
>
> - **TabArena does not allow comparing runtimes, which is critical for our project.** That is, to build a version of Figure 5 for TabArena, we would have to rerun all TabArena baselines on all TabArena datasets on our hardware, which is infeasible.
> - **We focus on middle-to-large datasets, while TabArena covers small-to-medium datasets**. In other words, we have all the ambitions to perform well on middle-to-large datasets, and a bit less so on smaller datasets. One of the reasons is the limitation of TabPack mentioned in Appendix C: having large enough validation sets may be important for our method, and this does not seem to be the case on many (especially small) TabArena datasets.

---

> > ### Author Rebuttal · Reviewer_91be · 2026-04-04
> >
> > I thank the authors for addressing several of my concerns.
> >
> > The rebuttal clarifies the intended scope of the paper, focusing on runtime-aware performance of non-foundation tabular deep learning models for medium-to-large datasets. While this positioning is internally consistent, it also highlights a key limitation: the empirical evaluation remains restricted to a specific regime and does not fully establish how the method compares to broader classes of modern approaches.
> >
> > In particular, while the rebuttal includes comparisons to tabular foundation models, these models are evaluated in settings in which they are known not to perform best (e.g., large-scale datasets). As a result, it remains unclear how competitive the method is in a more general tabular learning setting.
> >
> > Overall, my assessment remains unchanged.

---

> > > ### Author Response · Authors · 2026-04-07
> > >
> > > We thank the reviewer for the follow-up, and we appreciate the overall positive spirit of the review. For our part, we would like to complete the discussion by sharing our perspective on the remaining topics.
> > >
> > > The reviewer expresses interest in a specific evaluation regime (runtime-unaware performance evaluation) and in a specific type of models (foundation models). As researchers in the field of tabular ML, we fully respect and actually _share_ those interests. However, our _current_ study happens to focus on a _different_ evaluation regime (runtime-aware performance evaluation), and on the scope (middle-to-large datasets) where _different_ (non-foundation) models are more competitive today. And, in light of the latest reviewer's post, we would like to highlight the following:
> > >
> > > **Evaluating a study outside of its scope and evaluation regimes introduces a risk of dismissing positive contributions.**
> > >
> > > Let's use our paper as an example:
> > >
> > > - We present what may be the first non-foundation tabular DL model that delivers strong performance in a single run without hyperparameter tuning, which is a major shift for non-foundation tabular DL.
> > > - In particular, it makes experiment cycles _significantly_ faster. Consider this: **the main TabPack experiments can be conducted on a modern MacBook without using a discrete GPU at all** (see the details at the end of this post).
> > > - With the efficiency aspect in mind, all this indicates a big step forward for non-foundation tabular DL. At the same time, the same contributions are impossible to appreciate through the lens of runtime-unaware performance comparison with TFMs and/or on TabArena. Moreover, using that lens as the main one suddenly makes our study look "restricted" (on which we disagree).
> > >
> > > Another example is TFMs themselves (to state upfront: this is an imperfect analogy, and we do not draw direct parallels between those studies and ours). When they only appeared, they had significant limitations (e.g., they rarely performed well beyond small datasets). To the great benefit of the community, they found their way to the scene, despite lacking comparison with XGBoost in large-scale setups right from the start.
> > >
> > > Overall, again, we thank the reviewer for the discussion. We hope that our write-up will help us all to find the common ground.
> > >
> > > ---
> > >
> > > _**Bonus part. Training TabPack on a MacBook**_
> > >
> > > As mentioned above, the experiment runs of the $\mathrm{TabPack^\dagger}$ model from Figure 5 could be conducted locally on a MacBook in a reasonable time. As a quick illustration, the following table shows runtimes from the middle part of Figure 5 on the largest dataset used in the paper (723K objects and 136 features) for TabPack and TabM:
> > >
> > > | Model                      | Device                      | Runtime |
> > > | :------------------------- | :-------------------------- | ------: |
> > > | $\mathrm{TabM}^\dagger$    | NVIDIA A100                 | 12h 17m |
> > > | $\mathrm{TabPack}^\dagger$ | Apple M4 Pro GPU (20 Cores) |  3h 46m |
> > > | $\mathrm{TabPack}^\dagger$ | NVIDIA A100                 |  0h 24m |
> > >
> > > Thus, TabPack makes modern tabular DL more accessible not only in terms of hyperparameter tuning, but also in terms of hardware requirements. We believe that this is an exciting practical outcome of our work.
> > >
> > > P.S. Perhaps we should include the TabPack-on-MacBook experiments in Figure 5 in the camera-ready version.

---

### Official Review · Reviewer_crf3 · 2026-03-17

**Soundness:** 2
**Presentation:** 2
**Significance:** 2
**Originality:** 2
**Overall Recommendation:** 4
**Confidence:** 3

**Summary:**

This work propose TabPack, which use on-the-fly ensembling of same-family MLP models to produce ensembles in a single run. It has a relative good training efficiency and low tuning cost with such a design.

**Compliance With Llm Reviewing Policy:**

Affirmed.

**Final Justification:**

The rebuttal resolved my concerns.

**Key Questions For Authors:**

1. The ensemble cost is part of the performance improvement, which it is not totally clear whether it is taken into account.

There is also a confusing writing of the ensemble part:
As the ensemble is run every epoch and "on each iteration, add the next base model that improves the ensemble the most (on the validation set), without replacement", then if " multiple instances of same base model from different epochs can participate in the final ensemble.", why it is without replacement? And is the ensemble output-level?

And the ensemble method is not clear:
Is it average or majority vote or linear combination? As it is not clear, an ablation of it is also missing.

**Limitations:**

yes

**Strengths And Weaknesses:**

Strength:

1. The motivation is meaningful: packing tensor multiplications in gpu is much more efficient than running them separately.

2. The training is more time-efficient than tuning MLP baselines.

Weakness:

1. The overall relative inference cost is not clear. The authors said "For comparison, the ensemble size of TabM used in the original paper is 32, and TabM was shown to yield practical inference throughput with this ensemble size (Gorishniy et al., 2025). Hence, we conclude that the same is true for TabPack." However, inference cost needs to be estimated clearly in real world usage, so it is better to have a measurement rather than an ambiguous speculative "practical inference throughput"

2. High dependence on the validation set. The on-the-fly ensemble and early stopping highly relies on the verification set.

3. The framework is not plug-and-play enough. When new architecture elements or optimizers are introduced, they need to be adapted into packed version. (minor weakness, as LLM coding make it easier today)

4.Current base model is merely MLP (w/ w/o feature embeddings). This limit the ceiling of the performance.

5. Figure 6 comparison seems not fair enough, as the meta-parameters (num models and max width/depth etc.) of TabPack is tuned. Actually the hyper-parameter search space of TabPack leads to different fraction of usable base models.

---

> ### Author Rebuttal · Authors · 2026-03-31
>
> We thank the reviewer for the feedback, and we are looking forward to the discussion.
>
> ### Positive things
>
> We appreciate the reviewer's notes on the motivation behind our method's design, and especially on its efficiency, since this is a major focus of our work.
>
> ### Other topics
>
> > The overall relative inference cost is not clear ... it is better to have a measurement ... The ensemble cost ...
>
> The crucial detail is that **the inference-related measurements for supporting our claims already exist in prior work** `[1]`, and **we already refer to that paper** in L436(Left).
>
> Let us quickly summarize our reasoning:
>
> - **TabM is an ensemble of MLPs**,  and the TabM's inference throughput is already benchmarked in its original paper `[1]` (see the right side of Figure 4 and Figure 10 in `[1]`). The results show that **TabM is a practical solution**.
> - **TabPack is also an ensemble of MLPs, but with lower architectural hyperparameter values** (see the table below), which allows us to conclude that **TabPack is also a practical solution** without rerunning the whole inference benchmark.
>
> | | TabM `[1]` | TabPack (our work) |
> | --- | --- | --- |
> | Architecture | Ensemble of MLPs | Ensemble of MLPs |
> | Ensemble size in the original paper | $32$ | $15 \pm 9$ across all datasets (Max. value: $32$) |
> | MLP depth in the original paper | Tuned in $\mathrm{Uniform}[1, 5]$ | Varies in $[1, 3]$ |
> | MLP width in the original paper | Tuned in $\mathrm{Uniform}[64, 1024]$ | Fixed at $384$ |
>
> P.S. TabM and TabPack are not _exactly_ equivalent architectures, but the difference does _not_ affect the above reasoning in the context of inference throughput.
>
> > High dependence on the validation set.
>
> Please note that we already discuss this in our paper in Appendix C as a potential limitation. However, we do this mostly for transparency and out of caution for certain use cases, while our main results show that TabPack performs just fine in the scope of our benchmark (middle-to-large datasets).
>
> > The framework is not plug-and-play enough ... (minor weakness, as LLM coding make it easier today)
>
> As a quick note, this is also already discussed in our paper in Appendix C. Overall, we agree that this is a minor weakness, and we believe that the benefits of TabPack outweigh the implementation costs.
>
> > Current base model is merely MLP ... This limit the ceiling of the performance
>
> There are two important things to note:
>
> - **The main idea behind TabPack is not limited to MLPs**. At the same time, MLP is just enough for our work to present the main idea.
> - Furthermore, the recent literature on tabular DL suggests that **MLP is a representative tabular DL backbone:** many modern tabular DL architectures (e.g. ModernNCA, RealMLP, TabM) use simple variations of MLP as their backbones and outperform, say, attention-based architectures `[1]`.
>
> > Figure 6 comparison seems not fair enough, as the meta-parameters (num models and max width/depth etc.) of TabPack is tuned.
>
> **This comment seems to misinterpret Figure 6.** To clarify:
>
> - For TabPack, Figure 6 shows the dynamic of _one_ training run of TabPack with _fixed_ hyperparameters, so _no tuning is involved_. As we write in L251(right) in the paper: _"we never tune TabPack’s hyperparameters"_.
> - For baselines, Figure 6 shows the dynamic of hyperparameter tuning.
> - This is fair, because the main point here is that TabPack _does not need_ hyperparameter tuning, while baselines _need_ it.
>
> > confusing writing of the ensemble part ... why it is without replacement?
>
> We thank the reviewer for the question, and we will improve the ensemble algorithm description. To clarify:
>
> - After each epoch, a new ensemble is formed from scratch by greedy selection of base models from a pool of candidates _without replacement_.
> - At a given epoch, a pool of candidates is formed as "current base models" plus "members of the previous ensemble". Thus, for some models, multiple _different_ checkpoints (from _different_ epochs) are presented in the pool of candidates.
> - The greedy selection algorithm treats different checkpoints of the same model as "different" models. Thus, even though it operates in the "without replacement" regime, _"multiple instances of **same base model** from **different epochs** can participate in the final ensemble."_
>
> > is the ensemble output-level? ... Is it average or majority vote or linear combination?
>
> **The ensemble prediction is the mean prediction of its members.** The averaging is performed in the raw output space on regression tasks, and in the probability space on classification tasks.
>
> > an ablation of it is also missing.
>
> At early stages of the project, we tried various strategies, including linear combinations, but none of them was promising enough to justify the extra complexity. We consider those early attempts as "ideas that did not work", so we are hesitant to discuss them in the paper
>
> ---
>
> - `[1]` TabM: Advancing Tabular Deep Learning with Parameter-Efficient Ensembling

---

> > ### Author Rebuttal · Reviewer_crf3 · 2026-04-05
> >
> > I was questioning on ensemble cost because of the way the author trying to present it is not considered a standard way. The logic the author trying to claim is "because previous work did benchmarking, our theoretical cost is better, so we don't need to do it". But practical or not depends on real scenario, not just previous work's claim. We need quantitative data instead of just qualitative statement (no one can easily make a qualitative statement for all scenarios).
> > I would like to emphasise that a research work should be self-contained enough, so that the readers don't need to refer to previous work and take a guess. **A full benchmarking is not that necessary, but a basic measurement is helpful for real application scenario to make the trade-off decision.**
> >
> > I will remain my score until there is a direct response.

---

> > > ### Author Response · Authors · 2026-04-06
> > >
> > > We thank the reviewer for clarifying the request, and we are happy to provide specific measurements and to include them in the paper. Plus, at the end of this post, we share an efficiency-related bonus story.
> > >
> > > As suggested, we did not rerun the full benchmarking, but performed some basic measurements to quickly test our theoretical expectations. We consider the following setup (inspired by a similar experiment from the right part of Figure 10 of the TabM paper):
> > >
> > > - **All datasets of our benchmark** to ensure diversity across input dimensions (the number of features) and feature types (continuous, categorical). See Table 2 of our paper for dataset properties.
> > > - **The tuned models** for which we report the task performance at the left part of Figure 5, which corresponds to the real scenario of models optimized for task performance.
> > > - **NVIDIA A100 80GB & Batch size = 4096**, which corresponds to the real scenario of offline batched predictions.
> > >
> > > The table below reports the mean and standard deviations of the inference throughput aggregated across all datasets. The results show that **the measurements are in line with expectations.** Namely, on average:
> > >
> > > - A single MLP is the fastest model.
> > > - TabPack is faster than TabM.
> > > - Foundation models are significantly slower than the rest.
> > >
> > > | Model                           | Inference throughput (objects per second) |
> > > | :------------------------------ | :---------------------------------------: |
> > > | $\mathrm{TabICL}$-$\mathrm{V}2$ |                1K $\pm$ 1K                |
> > > | $\mathrm{ModernNCA^\dagger}$    |              323K $\pm$ 413K              |
> > > | $\mathrm{TabM^\dagger}$         |              464K $\pm$ 645K              |
> > > | $\mathrm{TabPack^\dagger}$      |              807K $\pm$ 768K              |
> > > | $\mathrm{MLP^\dagger}$          |               3.2M $\pm$ 2M               |
> > >
> > > Since aggregation inevitably results in information loss, we also provide a more informative visual presentation here:
> > > https://freeimage.host/i/throughput.B5HJDkg
> > >
> > >
> > > There are many more application scenarios with various hardware setups (CPU, GPU, TPU, XPU, ...), model hyperparameters (e.g., optimized specifically for inference), and numerous model-specific optimizations. Yet we hope that the new experiment, together with the theoretical grounding and insights from prior work, provides enough intuition to suggest that TabPack is applicable in real-world use cases.
> > >
> > > ---
> > >
> > > _**Bonus story. Training TabPack on a MacBook**_
> > >
> > > An additional fun fact that further shows that TabPack is indeed practical and efficient: the main TabPack experiments (i.e. $\mathrm{TabPack}^\dagger$ from Figure 5) can be conducted on a recent MacBook Pro in a reasonable time without using a discrete GPU at all.
> > >
> > > As a quick illustration, the following table shows runtimes from the middle part of Figure 5 on the largest dataset used in the paper (723K objects and 136 features) for TabPack and TabM:
> > >
> > > | Model                      | Device                      | Runtime |
> > > | :------------------------- | :-------------------------- | ------: |
> > > | $\mathrm{TabM}^\dagger$    | NVIDIA A100                 | 12h 17m |
> > > | $\mathrm{TabPack}^\dagger$ | Apple M4 Pro GPU (20 Cores) |  3h 46m |
> > > | $\mathrm{TabPack}^\dagger$ | NVIDIA A100                 |  0h 24m |
> > >
> > > Thus, TabPack makes modern tabular DL more accessible not only in terms of hyperparameter tuning, but also in terms of hardware requirements. We believe that this is an exciting practical outcome of our work.
> > >
> > > P.S. Perhaps, we should add the TabPack-on-MacBook experiments to Figure 5 in the camera-ready version.

---

### Decision · Program_Chairs · 2026-04-30

**Decision:**

Accept (regular)

**Comment:**

The authors introduce an memory-efficient ensemble method, TabPack, for MLPs, which they apply to tabular datasets. Their method takes the packed ensemble framework, adapts this framework to handle models with different hyperparameters, and includes a greedy model selection step during training. Though similar in flavour to recent methods like TabM, the authors achieve promising results on benchmarks. Strengths of TabPack include (1) its efficiency of ensemble training (though this benefit is shared by TabM), (2) the integration of hyperparameter selection into the ensembling procedure, and (3) the thoroughness of the evaluation in terms of accuracy, runtime, and ablations.

Nevertheless, there are several weaknesses in the existing paper:

- **Methodological limitations**: the training procedure relies heavily on a validation set for hyperparameter selection (though this limitation is noted by the authors). In addition, this method requires more memory than competing methods like TabM.
- **Limited scope**: the authors claim in their rebuttal that their setting is "the niche of non-foundation tabular DL models.” Admittedly, this is a narrow topic, and for a top-tier conference like ICML their paper would be better situated by increasing scope. Three ways to accomplish this are (1) testing small-to-medium sized datasets, (2) including TabArena results, even if a full comparison against baselines is infeasible, and (3) comparing against more baselines (see below).
- **Comparison against baseline methods**: the authors did not compare against tabular foundation models in the paper (though they added a TabICL comparison in the rebuttal). While the authors claim that many datasets would be OOM for other models like TabPFN 2.5, there are obvious ways to include these models through e.g. dataset subsampling. Given that all reviewers asked for a comparison against tabular foundation models to contextualize this method, it is fair to assume that most readers would want such a comparison and the authors should be more creative in finding ways to compare against these baselines.

Overall I believe this paper is suitable for ICML, but I would strongly encourage the authors to address the scope and comparison points below. Doing so will only benefit the paper and increase its chance for impact on the field.